# Deep learning based predictive modeling to screen natural compounds against TNF-alpha for the potential management of rheumatoid arthritis: Virtual screening to comprehensive in silico investigation

**Tasnia Nabi☯, Tanver Hasan Riyed☯, Akid Ornob✆ ***

Department of Biomedical Engineering, Military Institute of Science and Technology (MIST), Dhaka, Bangladesh

☯ These authors contributed equally to this work.

* akid.ornob@bme.mist.ac.bd

**Data Availability Statement:** All relevant data are within the manuscript and its Supporting information files. Additionally, the virtual screening

## Abstract

Rheumatoid arthritis (RA) affects an estimated 0.1% to 2.0% of the world's population, leading to a substantial impact on global health. The adverse effects and toxicity associated with conventional RA treatment pathways underscore the critical need to seek potential new therapeutic candidates, particularly those of natural sources that can treat the condition with minimal side effects. To address this challenge, this study employed a deep-learning (DL) based approach to conduct a virtual assessment of natural compounds against the Tumor Necrosis Factor-alpha (TNF-α) protein. TNF-α stands out as the primary pro-inflammatory cytokine, crucial in the development of RA. Our predictive model demonstrated appreciable performance, achieving MSE of 0.6, MAPE of 10%, and MAE of 0.5. The model was then deployed to screen a comprehensive set of 2563 natural compounds obtained from the Selleckchem database. Utilizing their predicted bioactivity ($pIC_{50}$), the top 128 compounds were identified. Among them, 68 compounds were taken for further analysis based on drug-likeness analysis. Subsequently, selected compounds underwent additional evaluation using molecular docking (< − 8.7 kcal/mol) and ADMET resulting in four compounds posing nominal toxicity, which were finally subjected to MD simulation for 200 ns. Later on, the stability of complexes was assessed via analysis encompassing RMSD, RMSF, Rg, H-Bonds, SASA, and Essential Dynamics. Ultimately, based on the total binding free energy estimated using the MM/GBSA method, Imperialine, Veratramine, and Gelsemine are proven to be potential natural inhibitors of TNF-α.

## Introduction

Rheumatoid arthritis (RA) is a complex autoimmune condition involving multiple genetic polymorphisms and is associated with a chronic inflammatory reaction in the joints and

dataset and the code for the deep learning model can be found at: https://github.com/nabitasnia/DL-model-and-Dataset.

**Funding:** The author(s) received no specific funding for this work.

**Competing interests:** The authors have declared that no competing interests exist.

probable association with destructive bone erosion [1]. The prevalence of this condition is estimated to impact between 0.1% to 2.0% of the global population [2]. RA's aetiology is mostly distinguished by synovial proliferation and inflammation, auto-antibody production, cartilage, and bone degradation. Tumor Necrosis Factor-alpha (TNF- α), Interleukin-1 (IL-1), and IL-6 are three key pro-inflammatory cytokines that are responsible for RA, among which TNF-α has been recognized as a crucial modulator of inflammatory reactions [3]. Secreted by Th1 cells and macrophages, TNF-α was initially thought to play a synergistic role in the development of RA [4] However, follow-up studies indicated that excessive activation of TNF-α signaling led to arthritis even in the absence of functional T and B cells [5]. Later studies found that even the membrane-bound form of TNF-α (mTNF-α) can lead to the full expression of arthritis [6, 7]. Additionally, synovial fibroblasts, activated by TNF-α, release pro-inflammatory cytokines such as IL-6, IL-1β which further accelerates bone erosion [7]. The multi-directional involvement of TNF- α in RA pathogenesis has made it a prime target for managing the disease's systemic effects as there is no recognized curative treatment currently. In recent years, a multifaceted treatment regimen has successfully decreased disease activity and alleviated the onset of systemic effects. Disease-modifying anti-rheumatic drugs (DMARDs), encompassing conventional synthetic, biologic, and targeted synthetic medications, demonstrate promising results but are associated with adverse effects. Biological DMARDs are expensive and impose a significant financial burden on the patients [8]. Additionally, conventional synthetic DMARDs, for instance, methotrexate, are associated with significant side effects, including infections, malignancies, hepatotoxicity, and teratogenesis in pregnancy [9]. Conventional non-steroidal anti-inflammatory drugs (NSAIDs) like naproxen and celecoxib can provide temporary pain relief but fall short in slowing disease progression [10].

The continuous progress in medical research has resulted in the acquisition of several bioactive molecules derived from medicinal plants that exhibit enhanced effectiveness and safety characteristics [11] against TNF-α. Triptolide, a diterpenoid epoxide produced by the plant *Tripterygium wilfordii Hook F*, demonstrated significant pharmacological effects particularly immunosuppressive impacts in the treatment of RA [12]. Another study focused on *Paeonia lactiflora Pallas* for its anti-inflammatory, palliative, and immune system-enhancing properties, utilizing the isolated total glycoside of paeony (TGP) from its roots [13]. The authors observed a therapeutic response in 71.7 percent of TGP-treated patients. Various other molecules such as *Aloe barbadensis*, *Alstonia scholarisa*, *Berberis lycium Royle*, *Cinnamomum verum*, *Citrus limon*, *Coriander sativum*, *Curcuma longathe*, *etc.* have been used as herbal interventions for the management of RA [14]. Natural substances also have drawbacks. Triptolide modulates immune cell activity and cytokine expression, but it has been associated with hepatotoxicity, nephrotoxicity, reproductive system toxicity, cardiotoxicity, splenic toxicity, pulmonary toxicity, and gastrointestinal tract toxicity [15]. Additionally, the quest for natural compounds with enhanced efficacy and assured safety has predominantly relied on manual methods, particularly literature reviews. This approach has overlooked many natural drug candidates that can potentially treat RA having minimal toxicity. Unfortunately, these candidates are not studied in a structured manner, limiting their potential application in clinical cases. In summary, manual screening limits the discovery of entire sets of natural ligands or inhibitors against specific proteins or receptors.

Virtual screening techniques, computational assessment of drug molecules against a particular target based on structural similarity, are currently used in the industry for rapid drug discovery [16]. More recently, a ligand-derived active pharmacophore model against TNF-α was generated to screen an in-house database consisting of 10,000 synthetic and natural compounds [17]. The authors identified five compounds based on computational screening and narrowed them down to three potential inhibitors after in-vitro validation and cytotoxicity

assays. However, the authors determined the pharmacophore structure based on only 26 known ligands against TNF-α. This small number may preclude important structural features from the model and prevent an exhaustive search for all possible inhibitors. Additionally, their work was not tailored exclusively to plant-derived ligands as their database contained a mixture of synthetic and natural compounds. Deep learning (DL) based virtual screening models can handle large volumes of structural data and can provide more accurate predictions than the conventional pharmacophore technique. A virtual screening approach to precisely find natural therapeutic agents against the SARS-CoV-2 main protease (Mpro) protein through Selleckchem was established using a DL model [18]. The study identified two compounds, Palmatine and Sauchinone, as promising inhibitors of Mpro protein. Such methodologies can be adapted to find a comprehensive list of plant-derived bioactive molecules targeting TNF-α. This paper presents a DL-based novel computational screening methodology to assess the potential of natural ligands in inhibiting the TNF-α protein. We deployed our model on the Selleckchem database which is a curated natural product library. The best performing TNF-α inhibitors are validated through binding affinity analysis and Molecular Dynamics (MD) simulation studies. A complete workflow is described in Fig 1. Moreover, the DL-based virtual screening approach can be integrated with in-vivo and in-vitro assays to create a complete drug discovery pipeline for any protein/target of interest.

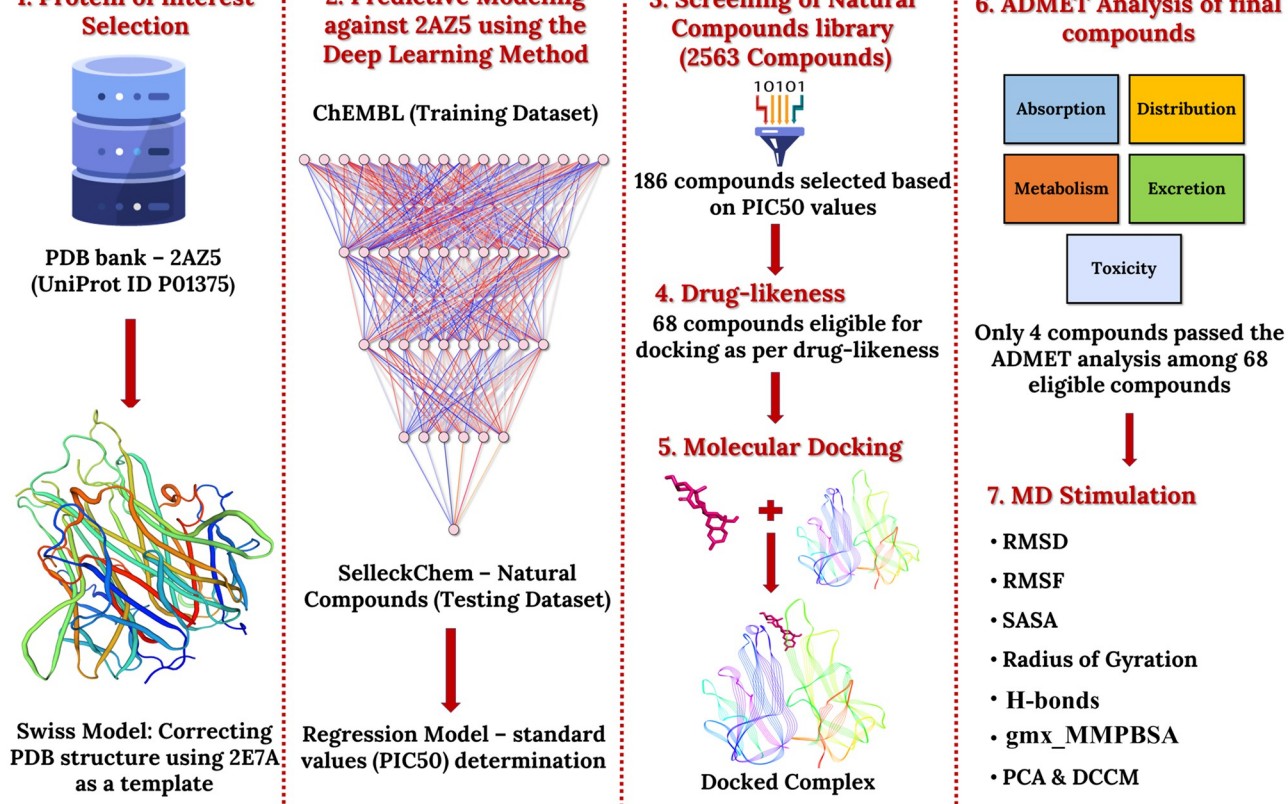

**Fig 1. Framework for DL-based virtual screening and predictive modeling of natural TNF-α inhibitors.** The model was developed using the ChEMBL dataset, with 2AZ5 as the target protein. After validation, the model was used to screen the Selleckchem Natural Compounds database. Compounds were prioritized based on their predicted $pIC_{50}$ values, and after additional filtering, four compounds were chosen for MD simulation studies.

## Materials and methods

### Refining target protein—TNF-α (2AZ5)

TNF-α's three-dimensional conformation was obtained from the Protein Data Bank [19]. The PDB ID: 2AZ5 structure of the TNF-alpha protein has been extensively studied in literature for different virtual screening studies [20]. The structure is complexed with a small molecule inhibitor and has become a benchmark for drug discovery studies aiming to inhibit the inflammatory activity of the TNF-alpha protein. The structure [21] was found to be incomplete due to the absence of protein residues which is summarized in S1 Table and visually depicted in S1 Fig. PDBsum [22] and the FirstGlance in the Jmol web server [23] aided in visualizing the absent regions. The SWISS-MODEL server [24] was utilized for homology modeling. This method utilizes the structural similarity of a closely related protein to predict the 3D structure of an unspecified protein. The FASTA sequence of 2AZ5 was uploaded to the server and the template search resulted in the identification of 2E7A [25], which displayed the highest similarity to the structure of 2AZ5. Furthermore, it outperformed other protein models in terms of evaluation metrics provided by SWISS-MODEL. The resulting model had GMQE score of 0.87, QMEANDisCO (Global) score of 0.82±0.05, and Ramachandran outlier of 0.46%, confirming the validity of its structure.

### Deep learning: Advanced predictive modeling

In our predictive DL model, ChEMBL ID-1825 [26] which corresponded to compounds against our target protein (2AZ5), served as a training dataset and contained 953 compounds in total. The dataset underwent filtration based on protein compatibility and reported $IC_{50}$ values, resulting in the retention of compounds having a set of unique canonical SMILES [27]. A negative logarithmic conversion was used to create a more uniformly distributed collection of $IC_{50}$ data, resulting in $pIC_{50}$. The PaDEL software [28] was employed to generate PubChem fingerprints [29] from the canonical SMILES. PubChem fingerprints contained 881 binary descriptors which indicate the existence of specific chemical compound groups [30]. To enhance the training dataset, we implemented variance thresholding to retain essential descriptors and reduced the amount from 881 to 235 in the training dataset. The same preprocessing techniques were applied during testing on the Selleckchem library [31]. The variance thresholding on the test dataset resulted in 251 essential descriptors. To ensure data uniformity, the descriptors from the train and test datasets were merged, and compared, and the repeating ones were eliminated to preserve 342 binary descriptors. The data pre-processing pipeline in both the train and test cases is highlighted in Fig 2.

We used RandomizedSearchCV for hyperparameter tuning [32] to find the best combination of parameters that optimize the given performance metrics. Finally, the best model incorporating the following parameters mentioned in Table 1 was later deployed on the test dataset consisting of 2563 compounds and the same 342 binary descriptors.

We modified the model's structure by modifying hyperparameters including the number of neurons, activation functions, and initializers for each hidden layer to match the data's individual properties. In the hidden layer, certain activation functions are adept at managing non-linearities, whereas other different activation functions are more effective at handling negative values, while the output layer uses a linear function. Choosing a weight initializer and a learning rate of 0.0001 helped our model converge and train faster without sacrificing stability. The "Adam" optimizer effectively adjusted the weights of the neural network while training.

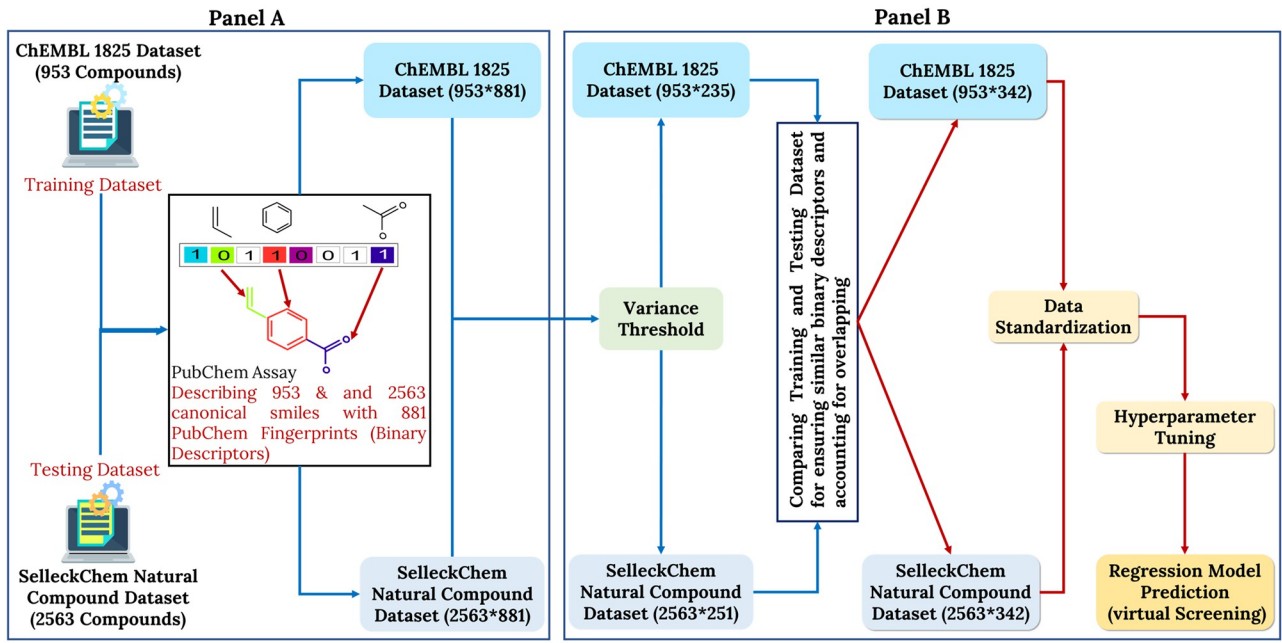

**Fig 2. Complete dataset pre-processing pipeline.** Panel A represents the characterization of compounds using PubChem descriptors (881) of both training (ChEMBL) and testing (Selleckchem) datasets and Panel B depicts variance thresholding to retain essential features.

## Optimizing models: Virtual screening assessment

The predictive model was assessed and reconstructed employing several statistical metrics. To evaluate the model's performance, we employed Mean squared error (MSE), Mean absolute percentage error (MAPE), and Mean absolute error (MAE) [33], which are employed to quantify the level of error present in statistical models. The regression model was utilized to analyze the Selleckchem natural product library, a subset of the Selleck database consisting of 2563 (July 2023) natural compounds, for virtual screening.

$$\text{MSE} = \frac{1}{k}\sum_{i=1}^{k}\left(y_i - \hat{y}_i\right)^2 \tag{1}$$

$$\text{MAE} = \frac{1}{k}\sum_{i=1}^{k}\left|y_i - \hat{y}_i\right| \tag{2}$$

$$\text{MAPE} = \frac{1}{k}\sum_{i=1}^{k}\left|\frac{y_i - \hat{y}_i}{y_i}\right| \times 100\% \tag{3}$$

**Table 1. Hyperparameter tuning of neural network architecture.**

| Number of hidden layers = 5 | Neurons | Activation | Initializer | Dropout |
|---|---|---|---|---|
| Unit 0 | 600 | tanh | he_normal | 0.1 |
| Unit 1 | 560 | relu | glorot_uniform | 0.1 |
| Unit 2 | 300 | relu | glorot_normal | 0.3 |
| Unit 3 | 420 | elu | he_normal | 0.2 |
| Unit 4 | 700 | tanh | uniform | 0.4 |

$k$ is the number of observations, $y_i$ represents the actual observed values, $\hat{y}_i$ represents the estimated values.

### Active site and binding site confirmation

The protein structure was analyzed using the CASTp [34] and DeepSite [35] web servers to determine its topological properties and predict the binding sites. This allowed us to identify and describe surface pockets and clefts, which are crucial for understanding the functional areas of the protein. These tools not only annotated the predicted binding pockets but also provided information regarding the involved interacting amino acid residues. The residues are represented in S2 and S3 Tables and S2 Fig.

### Drug likeness analysis

Following the virtual screening, the retrieved compounds underwent additional refinement by assessing their drug-likeness utilizing the SwissADME web server [36], which offers Lipinski filter, Veber filter, Ghose filter, Egan filter, and so on. These filters helped ensure that the selected compounds met specific criteria associated with drug-likeness [37].

### Molecular docking of screened compounds

Autodock Vina was utilized to conduct molecular docking between 68 screened compounds and the target protein [38]. Kollman charges and Polar hydrogen atoms were introduced into the 3D structure of the receptor during preprocessing. While merging non-polar hydrogen atoms in the ligand structures, the ligands underwent Gasteiger partial charges. To facilitate docking, a three-dimensional grid box was positioned at X = -4.173, Y = -6.686, and Z = 2.774 grid points. The grid spacing for the X, Y, and Z coordinates was set at 22 x 22 x 22 Å, respectively. For predicting the accurate result, the number of exhaustiveness was set to eight.

### ADMET analysis

The ADMET attributes, encompassing absorption, distribution, metabolism, excretion, and toxicity, were performed using the ADMETlab 2.0 [39] designed for systematic study of compounds' ADMET properties. It utilizes a vast ADMET database with over 280,000 entries. This platform provides support for the calculation of 88 ADMET-related endpoints across seven distinct categories.

### Retrospective validation of virtual screening protocol

Retrospective validation based on ligand shapes, fingerprints, and binding affinities have been reported in prior literature to distinguish between active ligands and decoy molecules [40, 41]. Previous reports have shown that validation based on 2D fingerprints similarity is fast and have shown similar or superior performance compared to the 3D shape based methods [42]. Therefore, we assessed our virtual screening using molecular fingerprints and utilized a subset of the DUD-E [41, 43] (Directory of Useful Decoys, Enhanced) database, which is widely recognized as a benchmark for evaluating virtual screening methodologies. To mitigate bias between active compounds and decoys, the decoys were selected based on similar physical properties but distinctive topological features. DUD-E is specifically designed to evaluate both structure-based and 3D ligand-based virtual screening techniques, with decoys deliberately chosen to have low 2D similarity to the active compounds [43]. We selected the top 20 active compounds for validation and generated 25 decoys for each of the selected active compounds.

The validation process employed two types of molecular fingerprint descriptors, Morgan and Layered fingerprints [44, 45], both calculated using the RDKit Python package. Morgan fingerprints were generated with a radius of 2 and a fingerprint size of 2048 bits, while Layered fingerprints were computed with a minimum path length of 1, a maximum path length of 9, and a fingerprint size of 2048 bits. For benchmarking purposes, we used 500 decoy molecules from the DUD-E dataset and included Schisantherin A [46], a natural TNF-alpha inhibitor, as a reference control and for internal thresholding. The anti-inflammatory activity of Schisantherin A against TNF-alpha is well-documented in both computational and in-vitro studies [47]. ROC plot showing the relationship between the True Positive Rate (TP) and False Positive Rate (FPR) was constructed for both the descriptors. Additionally, we calculated Enrichment Factor at top 1% based on fingerprint descriptors. All the active and decoy molecules also underwent molecular docking by the method outlined previously. The molecules were then ranked based on their binding affinity scores and the Enrichment Factor at top 1% was calculated to assess model performance in distinguishing between active and decoy compounds.

## MD simulation and post MD simulation

The GROMACS 23.1 [48, 49] package was utilized to perform molecular dynamics simulations, where CHARMM 36 force field [50] was applied to generate topology files utilizing the CGenFF server and pdb2gmx for both the ligand and protein. A dodecahedral system with periodic boundary conditions was constructed using box vectors of length (4.517, 6.309, 5.533 nm). The TIP3P water model was used to solvate the system and neutralize it by adding 4 positive sodium ($Na^+$) ions. Each solute within the box was positioned at the center, ensuring a minimum distance of 1 nm from the edge of the box. A collective total of 50,000 steps were taken by all systems during energy minimization and was conducted at 10 KJ/mol using the steepest descent algorithm with a Verlet cut-off scheme [51] and Particle Mesh Ewald (PME) [52] for Coulombic interactions. Subsequently, position restraints were applied during the equilibration step. The system underwent NVT equilibration and NPT equilibration at 300 K for 200 picoseconds (ps) [53]. Both systems maintained a constant number of particles (N) and temperature (T). The only distinction is that the volume of the NVT system remains constant, while the pressure of the NPT system remains constant. Finally, molecular dynamics simulation for apo-protein and protein-ligand complexes was executed for 200 nanoseconds (ns) with a time interval of 2 femtoseconds (fs). The generated trajectories were analyzed for root mean square deviation (RMSD), root mean square fluctuation (RMSF), radius of gyration (Rg), hydrogen bonding (H-Bonds), and solvent-accessible surface area (SASA), protein-ligand complexes using GROMACS applications.

## Principal component analysis (PCA) and dynamic cross-correlation matrix (DCCM)

The protein's functional activity was analyzed by studying its dynamic motions using Principal Component Analysis (PCA) [54] on the last 50 ns of MD simulation trajectories, using the Bio3d [55] package. PCA reduces the complexity of protein motion data by computing eigenvectors and eigenvalues. These are then used to identify and select different modes of motion, facilitating a more comprehensive understanding of the underlying dynamics [56]. Dynamic cross-correlation matrix (DCCM) [57] represents movements within a protein involving the coordinated shifts of α-carbon backbone atoms, influencing the transition of protein conformations between different functional states. Both PCA and DCCM analyses were executed on

aligned frames of C-α atoms from the respective MD trajectories, employing the fit.xyz function within the Bio3d package.

## Binding free energy calculation using gmx_MMGBSA

The computational method known as Molecular Mechanics Generalized Born Surface Area (MM/GBSA) [58, 59] is extensively employed in the estimation of the binding free energy of protein-ligand complexes. In this study, the gmx_MMGBSA [59] package was employed for conducting free energy calculations, utilizing a single trajectory generated by GROMACS with the CHARMM-36 force field. The evaluation was carried out using the following equation:

$$\Delta G_{bind} = \Delta G_{complex} - (\Delta G_{protein} + \Delta G_{ligand}) \tag{4}$$

$$\Delta G_{bind} = \Delta H - T\Delta S \tag{5}$$

$$\Delta H = \Delta E_{MM} + \Delta G_{solv} \tag{6}$$

$$\Delta E_{MM} = \Delta E_{bond} + \Delta E_{vdW} + \Delta E_{elec} \tag{7}$$

$$\Delta G_{solv} = \Delta G_B + \Delta G_{SA} \tag{8}$$

In Eq (4), $G_{Complex}$ represents the total binding energy of the complex, while $G_{receptor}$ and $G_{ligand}$ denote the energies associated with the receptor and ligand, respectively. Further elucidation of the changes in conformational entropy (–T$\Delta$S) and enthalpy ($\Delta$H) upon ligand binding is provided by additional Eqs (5–8). Here, $\Delta E_{MM}$, T$\Delta$S, and $\Delta G_{solv}$, represent alterations in gas-phase molecular mechanics energy, conformational entropy, and solvation free energy, respectively. Additionally, $\Delta E_{bond}$ stands for the energy of bonded interactions, assumed to be zero in dynamic simulations, while $\Delta E_{vdW}$, and $\Delta E_{elec}$ signify van der Waals and electrostatic interaction energies, respectively. Determination of the solvation free energy ($\Delta G_{solv}$) involves summing polar $\Delta G_{GB}$ (electrostatic solvation energy) and non-polar $\Delta G_{SA}$ between the solute and the continuum solvent. The non-polar component is assumed to be proportional to the molecule's solvent-accessible surface area (SASA). The impact of entropy ($\Delta$S) on the overall energy is negligible when comparing the binding states of ligands with the protein, hence it may be disregarded. Parameters utilized in this study include igb = 8, saltcon = 0.150, and intdiel = 8.

## Results

### Deep learning: Advancing predictive modeling

In the pursuit of constructing a robust regression model, we undertook an extensive approach utilizing PubChem fingerprint assays and implementing various preprocessing steps to establish standard values. To accurately assess DL models' performance and maintain their unbiased attitude toward the data, multiple evaluation metrics were implemented and yielded good performance with MSE (0.6), MAPE (10%), MAE (0.5), and the total time utilized for this model was 82.61 seconds. From Fig 3, it can be observed that there is little to less overfitting occurring between the training and testing set. The ChEMBL dataset was split into training and testing sets in an 80:20 ratio. Implementing dropout rates for each layer and hyper-parameterizing regularization terms: L1 (0.00113) and L2 (0.00015), proved effective in counteracting overfitting. The strategic application of the early stopping callback method, terminating training after 120 epochs, also helped in preventing overfitting by discerning the optimal point of

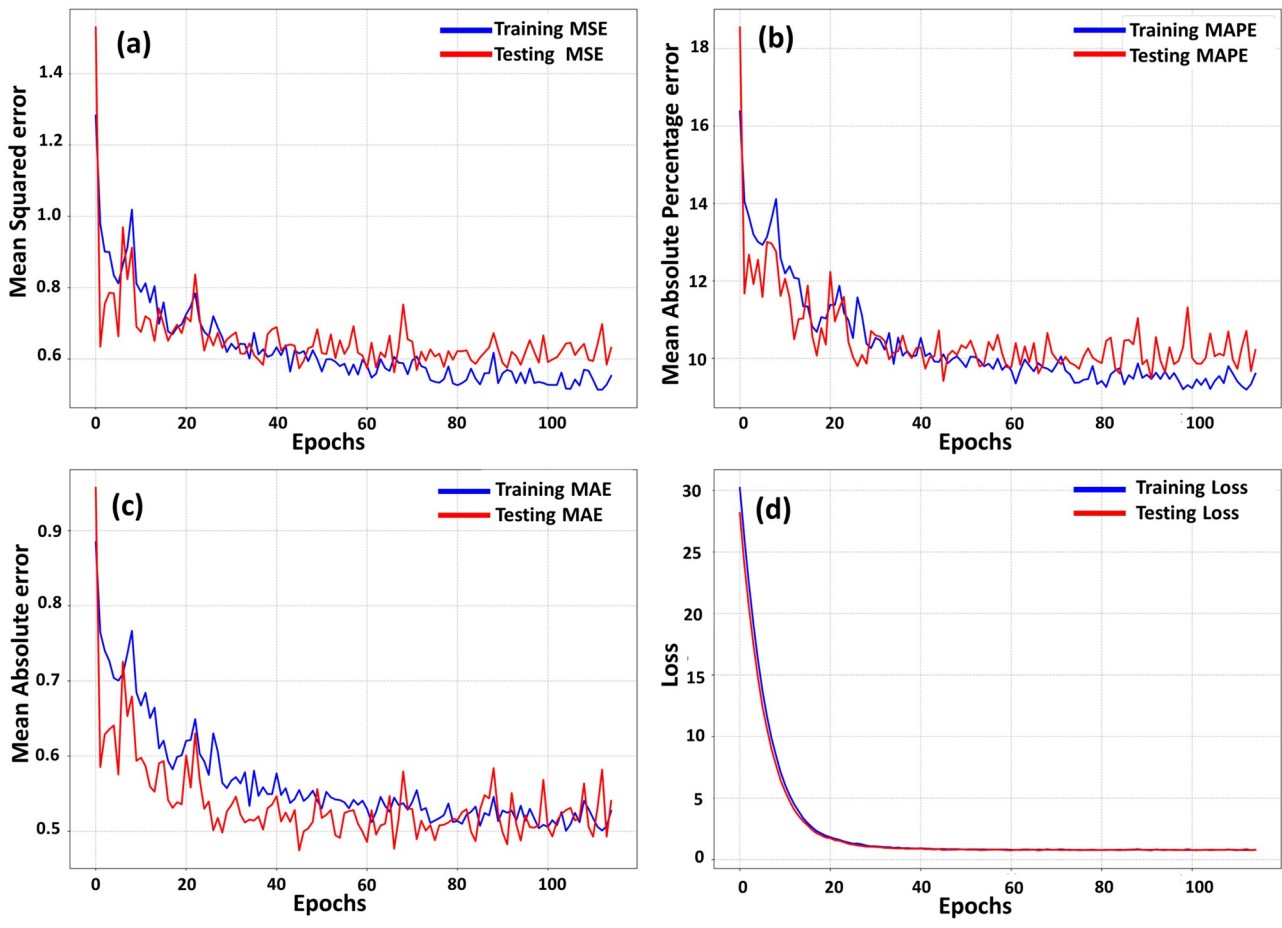

**Fig 3. Evaluation metrics of the DL regression model.** (a) MSE, (b) MAPE, (c) MAE, and (d) Loss.

model proficiency on the testing set. After this, the DL model was employed to screen a subset of the Selleckchem database, which encompasses 2563 natural compounds.

The predicted model yielded a mean predicted $pIC_{50}$ value of 5.276, with a range between 4.64 and 7.67. Compounds falling into the upper fourth quartile based on predicted values were exclusively selected. This criterion led to the inclusion of 186 compounds meeting the specified requirements. Despite their elevated $PIC_{50}$ values, the majority of these compounds were excluded due to their inadequacy in Drug-likeness analysis. As a result, only 68 compounds were chosen for subsequent and more thorough investigation. The structures of 12 of the predicted compounds are depicted in Fig 4 along with their $pIC_{50}$ values.

## Molecular docking

In the course of the investigation, molecular docking studies were conducted on 68 compounds, against our target protein resulting in binding energies between -10.4 and -5.1 kcal mol-1. Compounds chosen were docked into the active site of 2AZ5 utilizing AutoDock Vina, which predicts the optimal binding mode and reduced binding energy. Docking scores and $pIC_{50}$ values of 68 compounds are summarized in Fig 5.

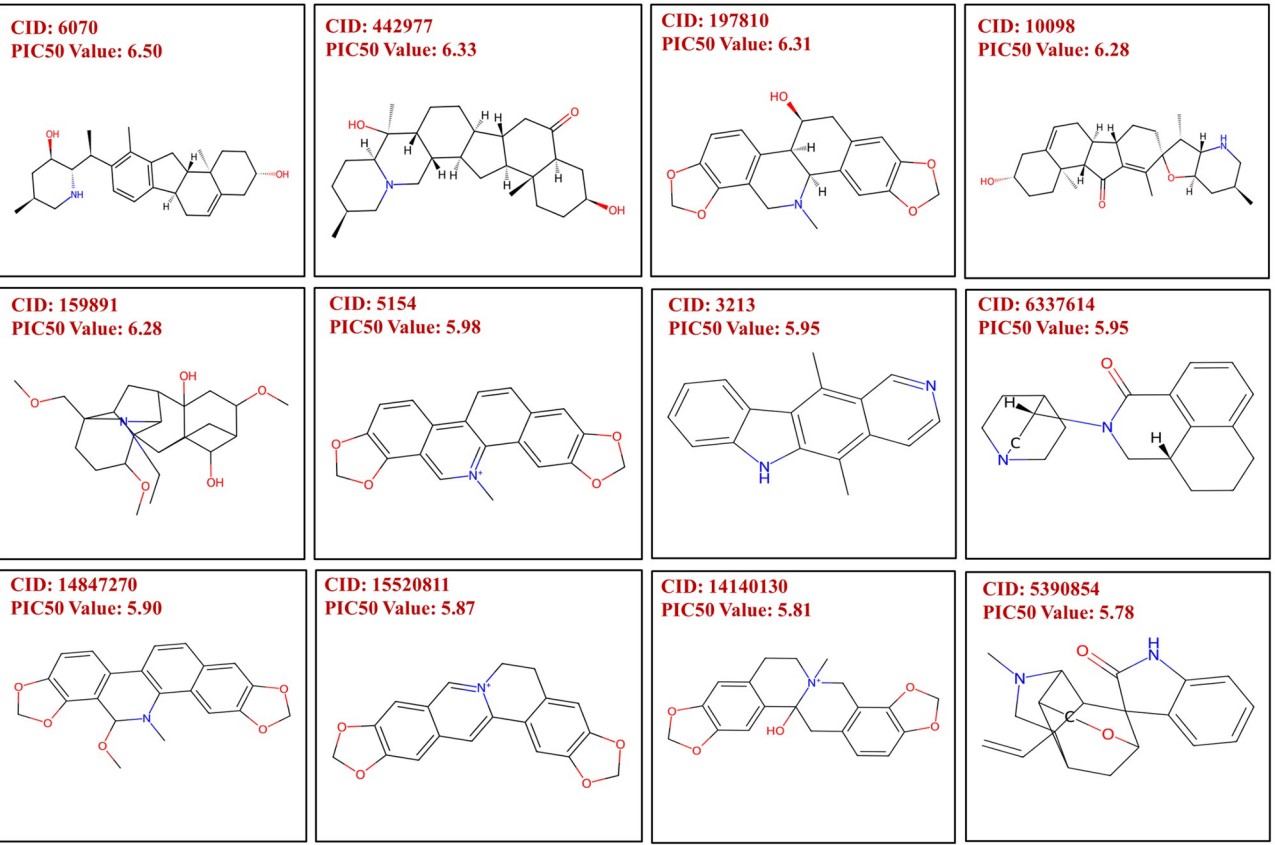

**Fig 4. Identification of screened compounds through Selleckchem's natural product library along with their predicted pIC$_{50}$ values.** The structures and the pIC$_{50}$ values of the eligible candidates are included (CID 442977: Imperialine, CID 6070: Veratramine, CID 10098: Jervine, and CID 5390854: Gelsemine).

## ADMET analysis and drug-likeness analysis of final 4 compounds

**ADME and toxicology properties.** High docking scores suggest a strong binding affinity between proteins and ligands. In total 23 Compounds were selected within the docking score range of -10.4 to -8.7 and subjected to ADMET analysis. Despite their high docking scores, a substantial proportion of the compounds exhibited harmful effects according to the ADMET evaluation. The identified cohort included only four compounds—Imperialine, Veratramine, Jervine, and Gelsemine—demonstrating commendable properties presented in Table 2.

In terms of absorption, all compounds were found to have better membrane permeability (Caco-2) around or greater than -5.15 log cm/s. Imperialine and Gelsemine act as P-gp substrates rather than P-gp inhibitors and contribute to the drug elimination process facilitated by enzymes. On the other hand, Veratramine and Jervine act as both P-gp inhibitors and substrates. The dual role of being both a substrate and an inhibitor may result in complex interactions with other drugs, affecting their bioavailability and therapeutic effects. In terms of Human Intestinal Absorption (HIA), molecules with HIA >30% were categorized as HIA- (Category 0) and those with HIA < 30% as HIA+ (Category 1). All our compounds belong to Category 0, signifying elevated HIA with values approaching zero. Drugs exerting their effects within the central nervous system (CNS) necessitate traversal of the blood–brain barrier to access their molecular targets. Apart from Gelsemine, the remaining three compounds

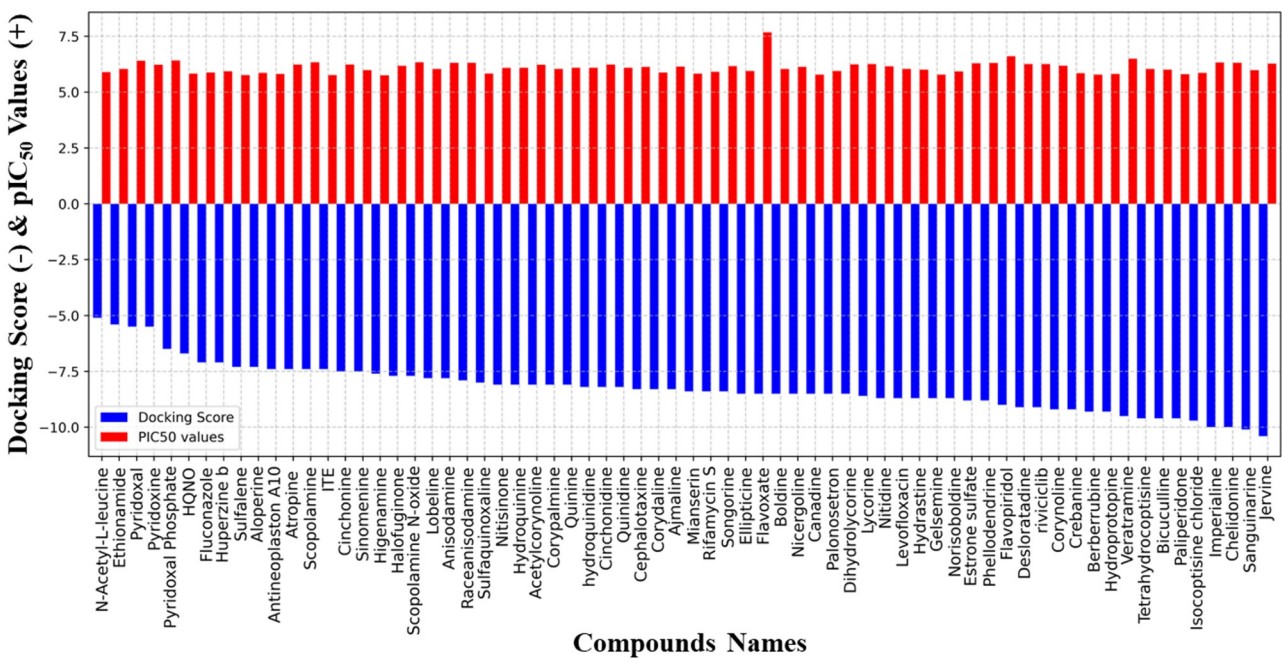

**Fig 5. Visualization of docking scores and pIC$_{50}$ values of 68 eligible natural candidates after drug-likeness analysis.**

**Table 2. ADME properties of the compounds Imperialine, Veratramine, Jervine, and Gelsemine using ADMETlab2.0 server.**

| Parameters | | Hit Compounds | | | |
|---|---|---|---|---|---|
| | | Imperialine | Veratramine | Jervine | Gelsemine |
| Absorption | Caco-2 permeability (log unit) | -4.769 | -5.066 | -4.882 | -5.443 |
| | MDCK | 2.20E-05 | 7.00E-06 | 1.40E-05 | 4.4E-05 |
| | P-glycoprotein Inhibitor | 0.215 | 0.967 | 0.969 | 0.003 |
| | P-glycoprotein Substrate | 0.999 | 0.992 | 0.973 | 0.801 |
| | HIA probability | 0.107 | 0.021 | 0.014 | 0.005 |
| Distribution | Plasma Protein Binding | 63.65% | 94.63% | 78.99% | 47.75% |
| | Blood-brain barrier (BBB) Penetration | 0.668 | 0.514 | 0.302 | 0.972 |
| Metabolism | CYP-1A2 inhibitor | 0.049 | 0.088 | 0.078 | 0.033 |
| | CYP-1A2 substrate | 0.239 | 0.688 | 0.587 | 0.959 |
| | CYP-2C19 inhibitor | 0.028 | 0.249 | 0.490 | 0.071 |
| | CYP-2C19 substrate | 0.854 | 0.941 | 0.911 | 0.944 |
| | CYP-2C9 inhibitor | 0.082 | 0.102 | 0.119 | 0.056 |
| | CYP-2C9 substrate | 0.043 | 0.187 | 0.033 | 0.094 |
| | CYP-2D6 inhibitor | 0.089 | 0.277 | 0.176 | 0.319 |
| | CYP-2D6 substrate | 0.758 | 0.889 | 0.842 | 0.830 |
| | CYP-3A4 inhibitor | 0.397 | 0.678 | 0.891 | 0.703 |
| | CYP-3A4 substrate | 0.528 | 0.406 | 0.525 | 0.920 |
| Excretion | Clearance | 18.09 | 10.59 | 9.99 | 13.37 |
| | T 1/2 | 0.229 | 0.074 | 0.136 | 0.149 |
| Toxicity | AMES Toxicity | 0.013 | 0.129 | 0.034 | 0.045 |
| | Carcinogenicity | 0.616 | 0.577 | 0.886 | 0.405 |
| | HERG- blockers | 0.372 | 0.866 | 0.457 | 0.419 |
| | H-HT | 0.377 | 0.318 | 0.388 | 0.296 |

exhibited satisfactory BBB values within the range of 0 to 0.7. Imperialine displayed an exceptional clearance rate exceeding 15 ml/min/kg, while veratramine, Jervine, and Gelsemine exhibited moderate clearance. Despite differences in clearance rates, all compounds showed a moderate half-life, reflecting a balanced pharmacokinetic profile with neither excessively rapid nor prolonged elimination.

**Toxicology.** From a quantitative perspective, four characteristics were predicted: hepatotoxicity (H-HT), carcinogenicity, AMES toxicity, and HERG blockers of acute toxicity. The hERG gene, responsible for encoding a potassium ion channel, is more strongly linked to severe cardiotoxicity. The AMES test is utilized to determine the mutagenic characteristics of a chemical, specifically its capacity to cause genetic harm and mutations. Upon comprehensive toxicity analysis, it was found that only the above four compounds exhibited favorable results across the majority of the evaluated properties, indicating a promising profile for further development. Conversely, those compounds that did not meet the established safety thresholds displayed multiple adverse toxicity indicators, such as Sanguinarine showcasing -10.1 kcal/mol binding affinity exhibited AMES toxicity, carcinogenicity along with respiratory toxicity, and many more, necessitating its exclusion from consideration. This rigorous selection process was crucial in ensuring the identification of potential candidates with the highest likelihood of safety and efficacy for further investigation and development as natural products.

**Drug likeness analysis.** The compounds that successfully passed ADMET analysis also showed promising drug-likeness results. Among the four compounds, Imperialine was reported to have more than 70 atoms, not adhering to the Ghose filter, but complying with the other four filters. However, all of our test compounds followed the Lipinski rule of five, ensuring proper absorption or permeation and thus qualifying as potential drug candidates against our target protein. The summary of Drug Likeness results is given in Table 3. The docked poses of these 4 compounds are represented in Fig 6.

In the context of protein-ligand interactions, it is apparent that Imperialine, Veratramine, Jervine, and Gelsemine exhibited binding affinity comprising the active site residues of 2AZ5. To gain a better understanding of the binding mechanisms of ADMET-approved compounds within the protein's active sites, we conducted a 2D interaction analysis for the docked complexes as depicted in Fig 7.

Pertaining to the protein-ligand complex, the equilibrium among various bond types, such as carbon-hydrogen bonds, alkyl bonds, and conventional hydrogen bonds, plays a pivotal role in influencing conformational dynamics, affinity, and biological function. Given the inherently weaker nature of carbon-hydrogen bonds in comparison to conventional hydrogen bonds, the stability of a ligand is anticipated to increase with a greater prevalence of conventional hydrogen bonds in the complex. Among the four selected drug candidates, Imperialine exhibited the formation of two conventional hydrogen bonds with ARG94 and LYS103, in conjunction with a carbon-hydrogen bond with CYS92. Additionally, it established four alkyl hydrophobic interactions with CYS92, PRO97, and ARG94, and thus contributed to enhanced structural flexibility. The 2AZ5-Veratramine complex displayed a diverse array of bond types, manifesting attractive charge interactions with GLU107 and amide-pi stacked bonds with PRO91. The pi-stacked interactions with the aromatic amino acid residues in the protein influenced the

**Table 3. Drug-likeness analysis of final four compounds.**

| Compound | Lipinski Rule | Ghose | Veber | Egan | Muegge |
|---|---|---|---|---|---|
| Imperialine | Yes | No | Yes | Yes | Yes |
| Veratramine | Yes | Yes | Yes | Yes | Yes |
| Jervine | Yes | Yes | Yes | Yes | Yes |
| Gelsemine | Yes | Yes | Yes | Yes | Yes |

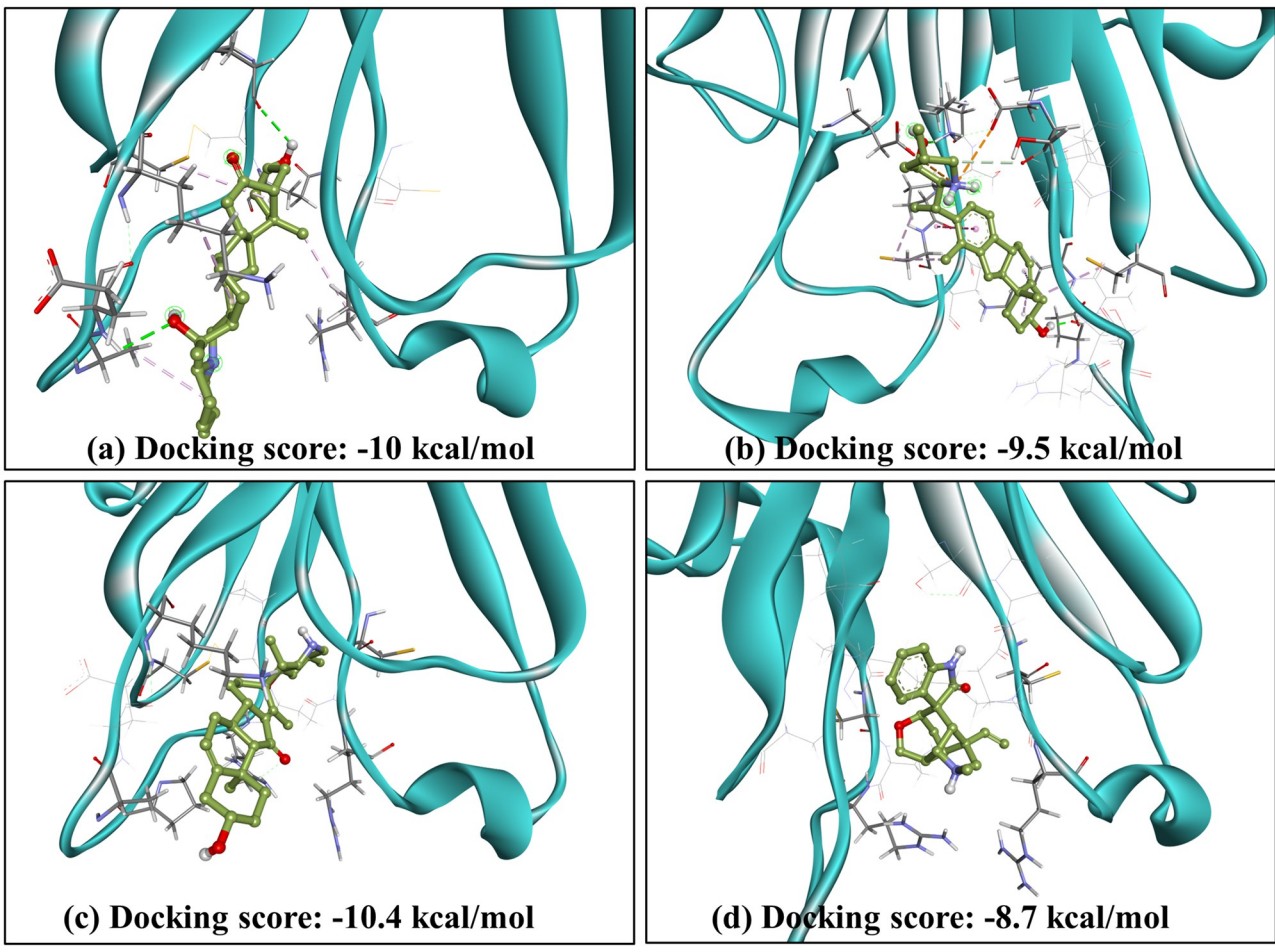

**Fig 6. Docked poses of top hit compounds against target TNF-α.** (a) Imperialine, (b) Veratramine, (c) Jervine, and (d) Gelsemine.

specificity of the binding. This complex further exhibited five alkyl bonds with CYS92, CYS60, and LYS103 which further stabilized the complex. Notably, two conventional hydrogen bonds with LYS89 and GLU101, along with a carbon-hydrogen bond with SER90, fortified the interaction profile. In the case of the 2AZ5-Jervine complex, alkyl-pi interactions, and a non-covalent bond mediated by van der Waals forces, contributed to heightened durability and prevented ligand dissociation. This complex was stabilized by one conventional hydrogen bond with GLU101, along with six alkyl and pi-alkyl hydrophobic interactions involving TRP105, CYS92, LYS103, and CYS60. Additionally, a carbon-hydrogen bond with CYS92 further enriched the interaction landscape. Conversely, the 2AZ5-Gelsemine complex diverged from the aforementioned patterns, featuring two carbon-hydrogen bonds with ARG94 and CYS92, without engaging in conventional hydrogen bonding. Alkyl hydrophobic interactions with CYS92 and ARG94, coupled with van der Waals interactions with TRP105, influenced protein conformation changes and overall dynamics of the complex.

### Retrospective validation

ROC curve analysis revealed a robust predictive performance for Morgan fingerprints, achieving an area under the curve (AUC) of 0.90, while Layered fingerprints demonstrated reasonable accuracy with an AUC of 0.80 as can be observed from Fig 8.

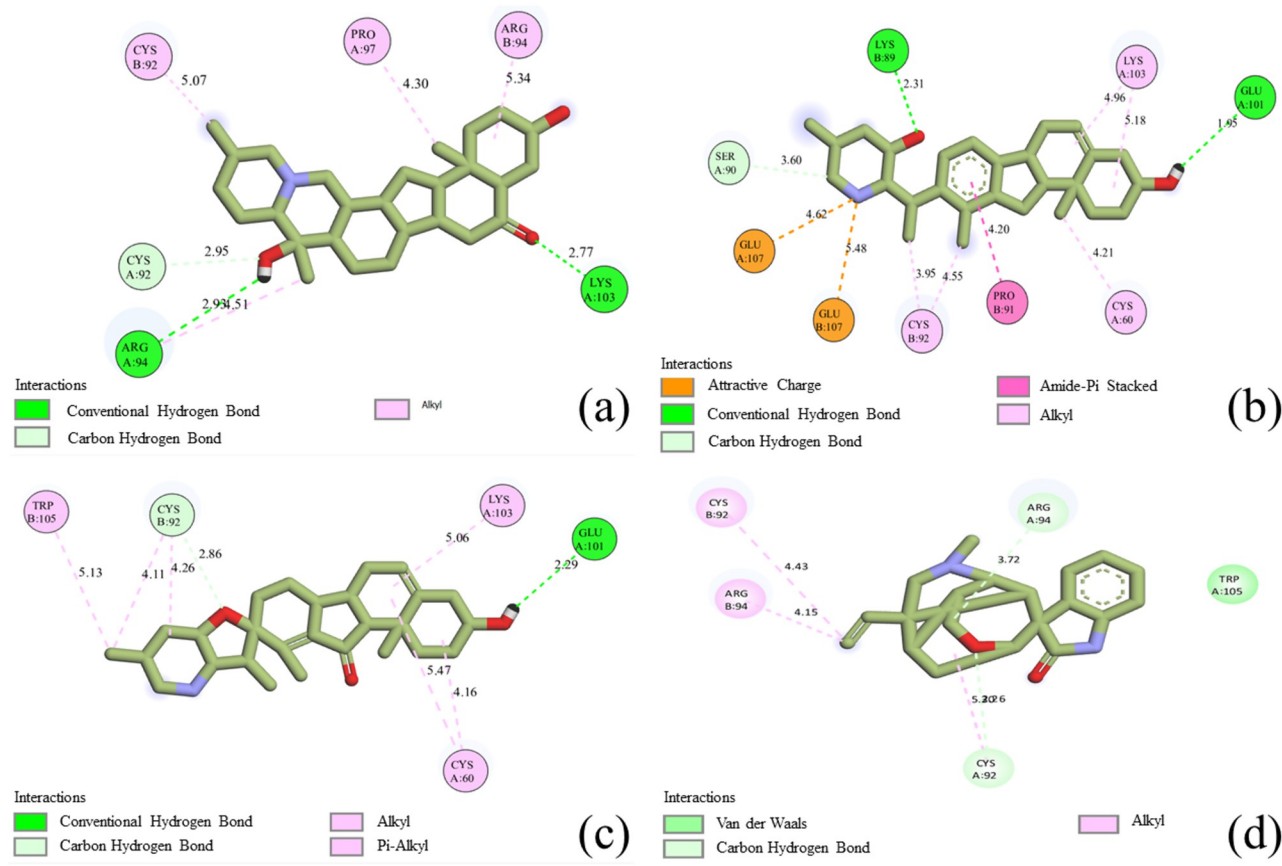

**Fig 7. 2D molecular interaction of top hit compounds.** (a) Imperialine, (b) Veratramine, (c) Jervine, and (d) Gelsemine.

The Enrichment Factor at 1% was 24.81 for Morgan and 12.40 for Layered, highlighting the model's ability to prioritize active compounds over decoys. The calculated Enrichment Factor (EF) at 1% for binding affinity was 18.18. Considering our limitations in using paid software and that the ROC-AUC metric may not always be optimal for evaluating binding affinity or docking scores in virtual screening, we used EF at top 1% to assess the performance of our model in prioritizing top-ranking compounds [60]. A similarity analysis using Tanimoto coefficients compared active compounds and decoys with Schisantherin A. For clarity, we visualized the first 50 compounds, starting with Schisantherin A as the control, followed by 20 active compounds and 29 decoys. The heatmap reveals a clear distinction between the control, active, and decoy compounds. The control compound (index 1) shows notable similarity with many of the active compounds (indices 2–21), while the active compounds themselves form clusters of moderate-to-high Tanimoto similarity, indicating shared structural features. In contrast, the decoys (indices 22–50) generally display lower similarity to both the control and active compounds, with only a few exceptions. Overall, the demarcations in the heat map indicate that active compounds are more similar to each other and the control, while the decoys remain largely dissimilar, validating the effectiveness of the virtual screening process. The boxplots demonstrate statistically significant differences in similarity values between active and decoy compounds for both Morgan ($p < 10^{-5}$) and Layered fingerprint ($p < 10^{-6}$) techniques. These results highlight that active compounds consistently show greater similarity than decoys, validating the effectiveness of the virtual screening process.

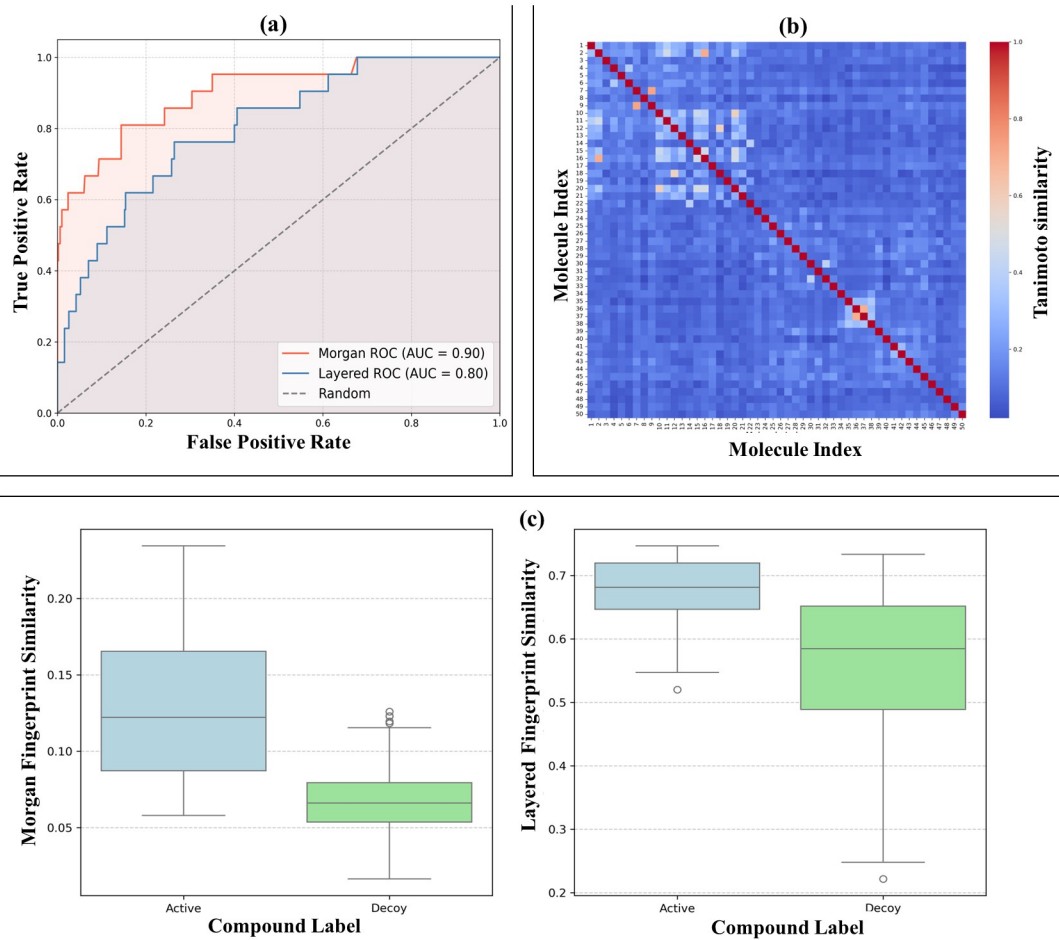

**Fig 8. Retrospective validation of our virtual screening protocol.** a) ROC plot based on Tanimoto scores for MF and Layered fingerprint descriptors b) Heat map showing Tanimoto similarity between control (index 1), actives (index 2–21) and decoys (22–50) and c) Box plots with distribution of Tanimoto scores between active-control and decoy-control for both the fingerprint descriptors.

## MD simulation

An exhaustive analysis of the MD trajectories of 2AZ5 was carried out, encompassing both its apoprotein state as well as its complex formation with four specific ligands over a simulation period of 200 ns. This facilitated the assessment of protein fluctuation and the influence of ligand binding on protein residues in these two states. It also aided in the identification of stable conformational sub-states and binding modes, as well as the detection of instabilities under dynamic circumstances to evaluate the potential of a natural ligand as a therapeutic candidate.

**Root mean square deviation (RMSD).** The RMSD values were computed for the complexes and 2AZ5 to evaluate the stability of each structure during the MD simulation. Overall, the four complexes exhibited low RMSD values corresponding to that of the apoprotein (Apoprotein: 0.254 ± 0.03 nm) demonstrated in Table 4. The time plot of the RMSD values for the protein backbone, shown in Fig 9, indicates that all four complexes exhibited fluctuations around 0.2 nm. While the RMSD values varied during the first 150 ns, they subsequently stabilized, indicating consistent behavior across the complexes. Notably, the ligand complexes— 2AZ5-Imperialine, 2AZ5-Veratramine, 2AZ5-Jervine, and 2AZ5-Gelsemine—demonstrated

**Table 4. Average structural and dynamic parameters for protein-ligand complex molecular dynamics simulation.**

| Protein And Protein-Ligand Complex | Average RMSD (nm) | Average RMSF (nm) | Average RG (nm) | Average SASA (nm) |
|---|---|---|---|---|
| Apo-Protein | 0.242±0.03 | 0.142±0.08 | 1.994±0.008 | - |
| Imperialine | 0.235±0.02 | 0.114±0.07 | 2.001±0.01 | 162.527±1.45 |
| Veratramine | 0.228±0.02 | 0.109±0.06 | 1.998±0.01 | 155.537±2.63 |
| Jervine | 0.236±0.03 | 0.109±0.08 | 1.991±0.01 | 164.457±1.51 |
| Gelsemine | 0.238±0.03 | 0.107±0.06 | 1.995±0.01 | 157.299±2.20 |

diminished fluctuations compared to the native protein, which is indicative of their potential inhibitory capabilities.

**Root mean square fluctuation (RMSF).** The complexes exhibited heightened stability relative to the unbound state. By observing the mobility of key residues of 2AZ5, it was visible that the critical residues at the binding site such as PRO91, CYS92, and LYS103 of chain A and SER90, PRO91, CYS92, GLN93 of chain B, had stable fluctuations with RMSF < 0.2 nm (Table 5). For the whole 200 ns simulation, the complexes exhibited lower fluctuations than the apoprotein in general. This was in full agreement with the RMSD values found earlier. Therefore, the results strongly suggested that protein-ligand complexes were stable at their binding site. Some atoms in these four complexes along with the apoprotein had high RMSF values but mostly exhibited values centered around 2 nm as observed in Fig 9(b). Certain active site residues with elevated RMSF may be naturally flexible or have undergone structural modifications, giving clues about conformational alterations linked to enzymatic activity.

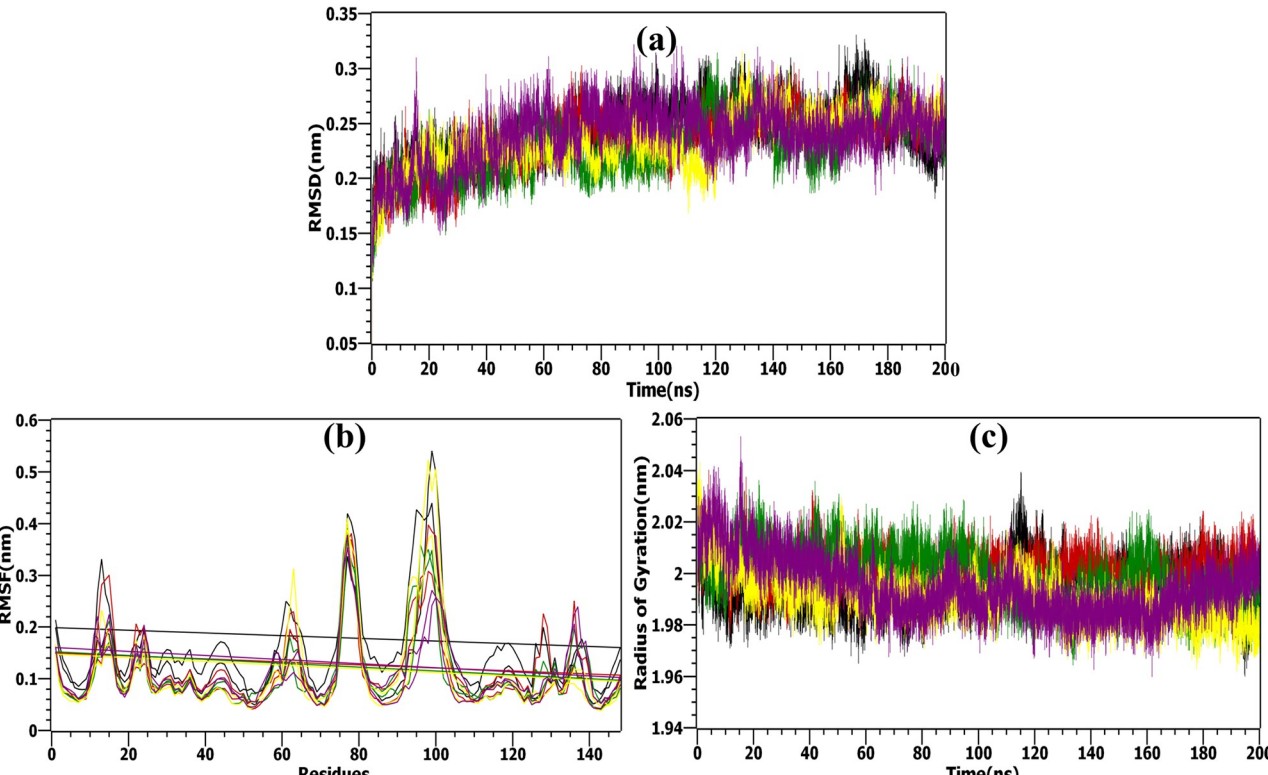

**Fig 9. MD simulation analysis.** (a) RMSD, (b) RMSF, (c) Rg values plotted for the native apo-protein -black and apo-protein docked with selected bioactive molecules, (1) Imperialine-red, (2) Veratramine-green, (3) Jervine-yellow, and (4) Gelsemine- purple.

**Table 5. RMSF (nm) of the active side residues for the apoprotein and the docked complexes.**

| Residues | Apoprotein | Imperialine | Veratramine | Jervine | Gelsemine |
|---|---|---|---|---|---|
| Chain A | | | | | |
| PRO91 | 0.146 | 0.121 | 0.122 | 0.135 | 0.121 |
| CYS92 | 0.195 | 0.143 | 0.107 | 0.156 | 0.101 |
| GLN93 | 0.287 | 0.221 | 0.245 | 0.277 | 0.156 |
| ARG94 | 0.353 | 0.251 | 0.279 | 0.297 | 0.156 |
| GLU95 | 0.427 | 0.258 | 0.279 | 0.294 | 0.147 |
| THR96 | 0.418 | 0.234 | 0.274 | 0.255 | 0.162 |
| PRO97 | 0.406 | 0.255 | 0.306 | 0.312 | 0.199 |
| ALA100 | 0.345 | 0.257 | 0.257 | 0.307 | 0.258 |
| LYS103 | 0.170 | 0.133 | 0.089 | 0.107 | 0.108 |
| Chain B | | | | | |
| SER90 | 0.082 | 0.063 | 0.070 | 0.060 | 0.055 |
| PRO91 | 0.094 | 0.073 | 0.080 | 0.074 | 0.068 |
| CYS92 | 0.099 | 0.100 | 0.085 | 0.080 | 0.087 |
| GLN93 | 0.150 | 0.161 | 0.091 | 0.099 | 0.080 |
| ARG94 | 0.150 | 0.180 | 0.137 | 0.170 | 0.094 |
| GLU95 | 0.198 | 0.188 | 0.216 | 0.254 | 0.112 |

**Radius of gyration (Rg).** The assessment of the compactness for all four complexes over the entire simulation duration was discerned by examining the Radius of Gyration (Rg) data, as illustrated in Fig 9(c). The Rg values exhibited a consistent trend throughout the MD simulation, indicating a persistently folded structure. As outlined in Table 4, it is evident that all the systems displayed comparable and uniform values of Rg suggesting compactness and stability throughout the entire simulation period.

**Hydrogen bonds (H-bonds).** Hydrogen bonds played a key role in maintaining the stability of the complex, with higher numbers of these bonds correlating with increased protein stability. Fig 10(a) illustrates the evolution of hydrogen bond formations over time. Throughout the simulation, the number of hydrogen bonds fluctuated between 2 and 9, reflecting changes in molecular interactions. The moderate level of hydrogen bonds observed in the complexes of Imperialine, Veratramine, and Gelsemine with 2AZ5 suggested that their hydrogen bonding networks remained stable during the simulation. In contrast, Jervine showed lower overall

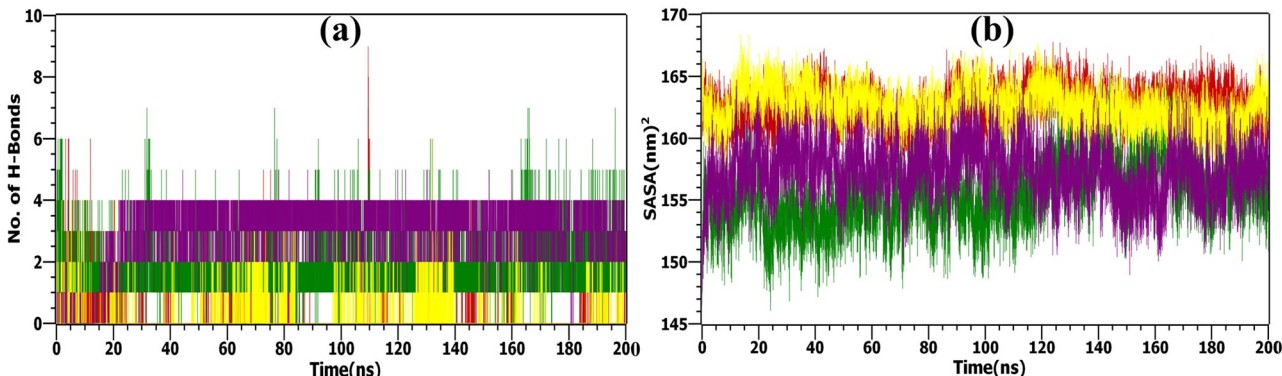

**Fig 10. MD simulation analysis.** (a) H-bonds, and (b) SASA values plotted for the native apo-protein (TNF-α in the human body)-black and apoprotein docked with selected bioactive molecules, (1) Imperialine-red, (2) Veratramine-green, (3) Jervine-yellow, and (4) Gelsemine–purple.

**Table 6. The calculation of binding free energy results of 4 selected compounds.**

| Complex | $\Delta E_{vdW}$ (kcal/mol) | $\Delta E_{elec}$ (kcal/mol) | $\Delta G_{GB}$ (kcal/mol) | $\Delta G_{SA}$ (kcal/mol) | $\Delta G_{gas}$ (kcal/mol) | $\Delta G_{sol}$ (kcal/mol) | $\Delta G_{bind}$ (kcal/mol) |
|---|---|---|---|---|---|---|---|
| Imperialine | -22.13 | -0.16 | 1.11 | -3.05 | -22.29 | -1.94 | -24.23 |
| Veratramine | -50.32 | -16.50 | 18.39 | -6.49 | -66.82 | 11.91 | -54.91 |
| Jervine | -7.74 | -0.52 | 0.96 | -1.06 | -8.25 | -0.10 | -8.35 |
| Gelsemine | -30.03 | -7.94 | 9.13 | -3.73 | -37.97 | 5.40 | -32.57 |

efficiency, consistently forming only two hydrogen bonds throughout most of the simulation, although there were brief moments where it formed up to six hydrogen bonds. A detailed examination of the intermolecular hydrogen bonds between the protein and ligand indicates sustained engagement throughout the entire 200 ns simulation, confirming the stability of these complexes.

**Solvent accessible surface area (SASA).**   Lastly, SASA analysis in Fig 10(b) demonstrated that all ligand complexes maintained relatively stable surface areas, reinforcing their enhanced stability. The binding of the ligand did not introduce any instability in the 3D structure as demonstrated by fluctuations around a stable value. The surface area of the complex can impact its stability by modulating the creation of water bridges (intermolecular hydrogen bonds) between the protein and the ligand. The results of the SASA are in good agreement with the binding energy calculations as Jervine demonstrated the maximum SASA also showed the least favorable binding energy values (Table 6). The opposite held for Veratramine which exhibited the lowest SASA and binding energy values.

## Essential dynamics and dynamic cross-correlation matrix analysis

The RMSD of the protein backbone stabilized at approximately 0.2 nm, indicating that the structure achieved relative stability after 150 ns during a 200 ns molecular simulation conducted at 300 K. Given these RMSD results, the final 50 ns of the simulation trajectory were utilized to calculate DCCM and to perform PCA.

## Principal component analysis

In 200 ns simulation trajectories, the top 20 PCs of the apo-system, 2AZ5-Imperialine, 2AZ5-Veratramine, 2AZ5-Jervine, and 2AZ5-Gelsemine system accounted for 84%, 74.3%, 77.3%, 71.6%, and 74.8% of the total variation, respectively. It was observed that the first few eigenvectors contributed mostly to elucidate the overall dynamics of the protein. The generated 2D plots indicated two distinct conformational states, denoted by blue (unstable) and red (stable) dots, with white dots representing intermediate states [61]. Fig 11 reveals that the percentage of variation in the 2AZ5 structure exhibited a significant decline corresponding to the first four eigenvectors, indicating the conformational changes caused by the docked ligand. Moreover, the eigenvalues for both docked complexes remained relatively constant from 5 to 20 eigenvectors, pointing to a lack of major changes within this range. These observations suggest that the 2AZ5 structure, when bound to the docked compounds, initially showed considerable flexibility, which then decreased over time. Notably, all systems demonstrated compact and correlated motions for the 2AZ5 structure, indicating a rigid and stable state for the docked complex.

## Dynamic cross-correlation matrix

DCCM captured the synchronized displacements of Cα backbone atoms that impact the transition of protein conformations across different functional states. It exhibited the collective

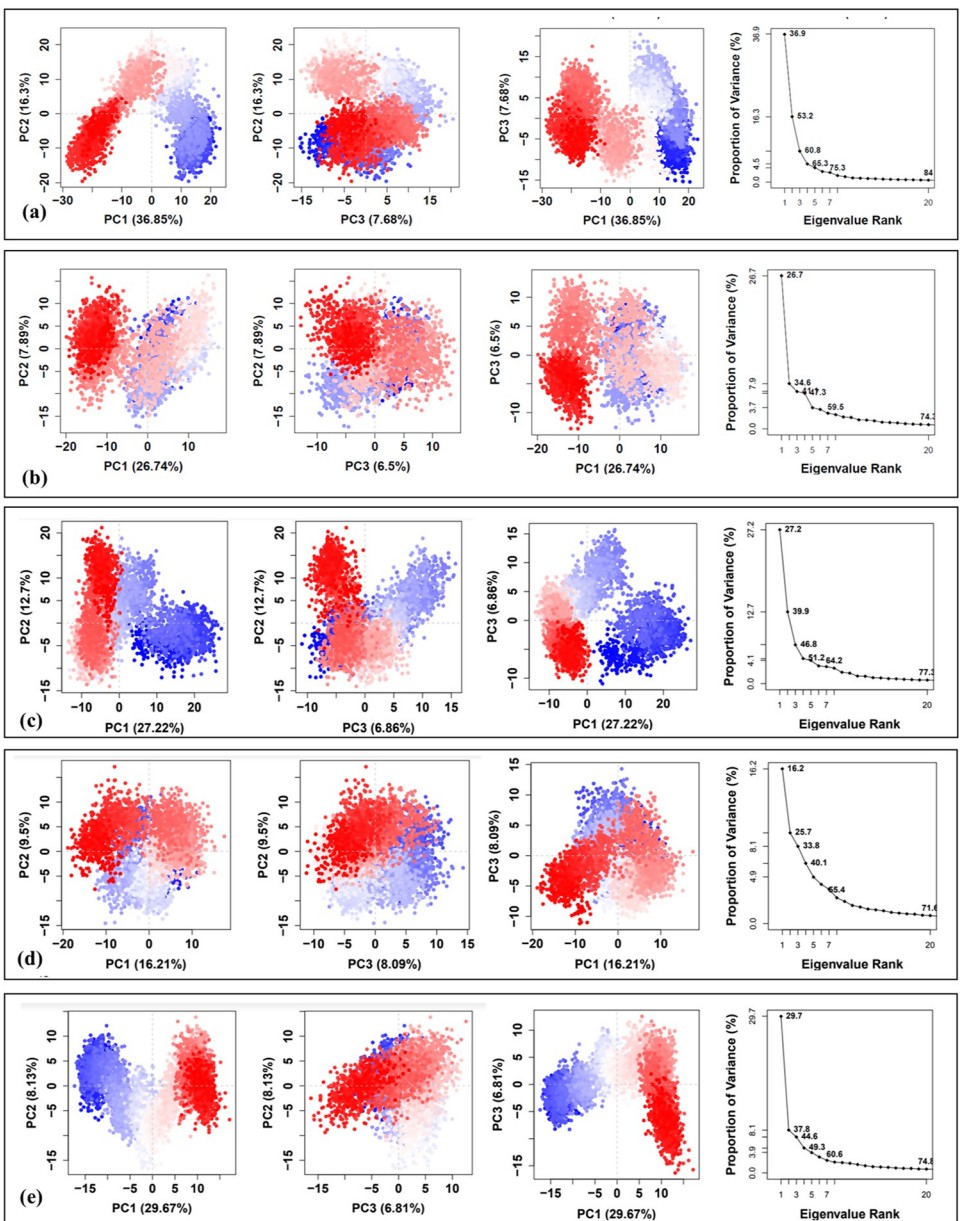

**Fig 11. Principal component analysis of the MD simulation trajectories.** (a) 2AZ5, docked with (b) Imperialine, (c) Veratramine, (d) Jervine, and (e) Gelsemine.

movement of Cα atoms and, by analyzing their relative motions, revealed the connections between residues and domains. Positive values within the matrix signified a favorable correlation, denoting the concerted movement of residues in a parallel direction. Conversely, negative values indicated anti-correlations, signaling the anti-parallel displacement of residues within complexes. In the accompanying figures (Fig 12), the intensity of the color reflected the strength of the correlation, with red representing a positive correlation, white denoting no correlation, and blue indicating a negative correlation.

Upon examining the DCCM of the apo-protein in Fig 12(a), a mostly negative correlation in the overall conformation was evident. This had been diminished notably in the docked

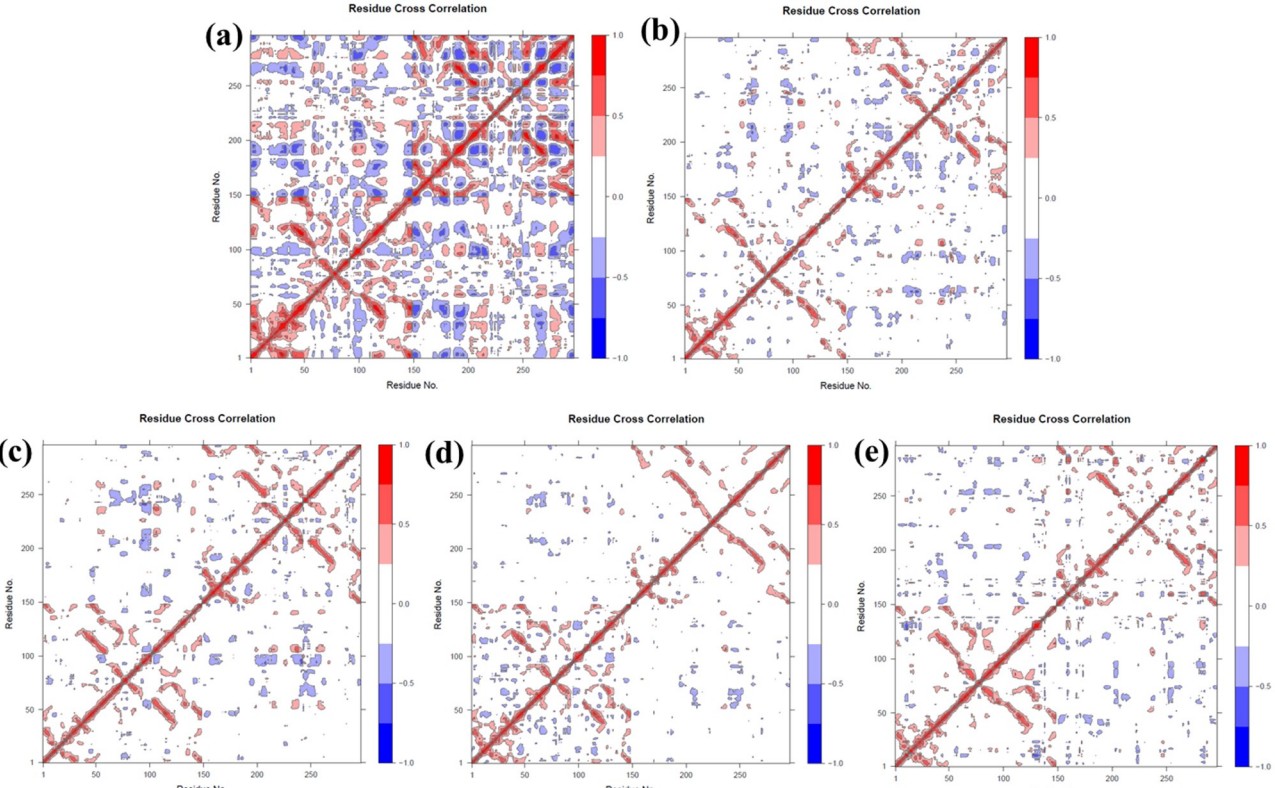

**Fig 12. Dynamic cross-correlation matrix of MD simulation trajectories.** 2AZ5 complexed with (a) 2AZ5 apo form, (b) Imperialine, (c) Veratramine, (d) Jervine, and (e) Gelsemine on α-carbon atoms.

complexes The shift in motion toward a slightly positive correlated motion or a lack of negative correlation implied a decrease in overall residue movement. This pointed to improved stability thereby leading to reduced conformational flexibility.

## MM/GBSA

The MM/GBSA method was used to estimate the total binding free energy ($\Delta G_{bind}$) for the complexes based on the last 10 ns of the simulation trajectory. The complexes containing Veratramine, Gelsemine, and Imperialine exhibited $\Delta G_{bind}$ values of –54.91 kcal/mol, –32.57 kcal/mol, and -24.23 kcal/mol, respectively. The observed low total binding free energy was attributed to the significant contributions from both $\Delta G_{gas}$ and $\Delta E_{vdW}$. Notably, the polar solvation energy was unable to make a positive contribution to the overall binding in any of the investigated complexes. Both electrostatic energy and non-polar solvation energy contributed similarly to the binding free energy. Conversely, Jervine displayed higher $\Delta G_{bind}$ values of—8.35 kcal/mol, indicating a weaker interaction with the 2AZ5 active site for these compounds. Through comprehensive analyses encompassing complex and ligand stability, as well as MM/GBSA binding free energies, Imperialine, Veratramine, and Gelsemine were identified as the most promising compounds.

## Discussion

We scrutinized multiple possible inhibitors of TNF-α from a wide range of natural substances of the Selleckchem database through molecular docking, drug-likeness, and ADMET analysis

process. Consequently, we have narrowed down our selection to four probable compounds: Imperialine, Veratramine, Jervine, and Gelsemine. Imperialine, from Bulbs of *Fritillaria cirrhosa*, is a steroidal alkaloid that has been scientifically proven to suppress inflammatory responses and mitigate pulmonary functional and structural impairment [62]. Additionally, Veratramine and Jervine, also steroidal alkaloids, have been recognized for their pharmacological activities. Among these, Veratramine has been reported to lower blood pressure and act as an anti-thrombotic agent [63]. Jervine, a constituent of *Veratrum album*, has demonstrated significant anti-inflammatory effects, ranging from 50.4% to 73.5%, against inflammation biomarkers such as TNF-α, IL-1β, and MPO [64]. Gelsemine, a natural alkaloid found in the *Gelsemium* genus of plants within the *Loganiaceae* family, has also been shown to possess anxiolytic effects and reduce the levels of pro-inflammatory cytokines [65]. Prior research has documented the use of natural alkaloids to yield promising outcomes in the management of inflammatory conditions such as RA. Therefore, the compounds identified through our screening process are well-validated and could emerge as promising therapeutic agents for treating RA and reducing its systemic symptoms.

Our investigation commenced with the selection of Protein ID 2AZ5 based on validation scores and recent literature references. Addressing missing residues, we aligned it with the template structure 2E7A to ensure accessibility. This technique can be extended to investigate an unknown protein of interest by aligning its structure with relatively matched ones from relevant databases. After the identification of structural homologs, the structural hits can be used to virtually screen databases containing potential inhibitors by integrating DL techniques. In a broader context, DL models enable comprehensive analyses across diverse databases of interest. Leveraging a curated dataset from the ChEMBL repository, we meticulously trained our model to discern intricate molecular features indicative of potential binding affinities by filtering the dataset based on their $pIC_{50}$ values. Our preprocessing steps ensured data standardization, reducing redundancy which was reflected in the excellent performance of our selected compounds as evidenced by our MD simulation, Essential Dynamics, and MM-GBSA analysis. Overall, our DL model demonstrated good performance and had MAPE value of 10%. This outcome can be further improved by using a larger training dataset or incorporating data from multiple different datasets. To ensure that the top hits were not accidentally eliminated, we selected the compounds falling in the upper fourth quartile after deploying our model on the Selleckchem database. This confirmed that the top hits were all investigated for detailed molecular docking studies, drug-likeness, and ADMET analysis. Additionally, when investigating any known or unknown protein rather than TNF-α, these stringent screening criteria may be altered, such as changing the variance threshold, the desired range of predicted values, Druglikeness properties, cut-off values of binding affinity, etc. to get the highest number of desired targets. Most importantly, our proposed virtual screening model is well validated as it was capable of successfully distinguishing between active ligands and decoy molecules based on two popular fingerprint descriptors.

Notably, our exploration extended beyond conventional docking analyses, encompassing Molecular Dynamics simulations to assess the dynamic behavior and stability of all four protein-ligand complexes over extended timeframes. These simulations provided valuable insights into the conformational dynamics of TNF-α and its interactions with potential inhibitors, shedding light on the mechanisms underlying inhibition. Furthermore, PCA & DCCM analysis offered a holistic view of the collective conformational changes induced by the selected compounds, illuminating their potential impact on TNF-α activity. In addition, complexes containing Imperialine, Veratramine, and Gelsemine exhibited remarkable results in binding energy analysis. Conversely, Jervine displayed weaker interactions with the protein active site, as indicated by their higher $\Delta G_{bind}$ values. Through comprehensive analyses encompassing

complex and ligand stability, as well as MM/GBSA binding free energies, Imperialine, Veratramine, and Gelsemine emerged as the most promising compounds for further investigation. Their exceptional binding affinities and favorable interaction profiles make them attractive candidates for therapeutic intervention in inflammatory diseases such as RA. As with any computational studies, our study aimed to optimize between computing power and cost. This optimization meant that we were only able to utilize a fraction of the 200 ns simulation for PCA, DCCM, and MM/GBSA studies. Overall, our study underscores the power of computational methodologies in accelerating drug discovery efforts, offering a systematic approach to identify promising candidates for therapeutic intervention, and contributing to the ongoing quest for effective treatments. Moving forward, our research sets the stage for further investigations, including experimental validation of identified compounds and exploration of their efficacy in preclinical models.

## Conclusion

In this study, we present a comprehensive and validated virtual screening pipeline that leverages deep learning techniques to rapidly search databases for natural compounds overcoming the limitation of manual screening and facilitating the exploration of potential drug candidates for RA. Our focus was on identifying natural compounds capable of inhibiting TNF-α, as potential therapeutic agents exhibiting superior safety and efficacy compared to existing drugs. From the approximately 2563 substances within the Selleckchem natural product library, our study employed a DL-based model for screening, followed by assessment through ADMET prediction and docking to initially identify four potential candidates: Imperialine, Veratramine, Jervine, and Gelsemine. Subsequent application of MD simulation and essential dynamics was conducted for these selected compounds. The MM/GBSA approach was then employed to identify top-hit compounds by calculating the binding capacity of each compound to the target, ultimately leading to the identification of Imperialine, Veratramine, and Gelsemine as the most promising TNF-α inhibitors.

## Supporting information

**S1 Fig. Missing amino acid representation among 4 chains of native 2AZ5 structure.** (TIFF)

**S2 Fig. Active site prediction using, (A) CASTp and (B) DeepSite.** (TIFF)

**S1 Table. Missing amino acid residues among 4 chains of native protein (TNF-α) and their position range.** (DOCX)

**S2 Table. Characterization of protein active sites using CASTp server.** (DOCX)

**S3 Table. Characterization of protein active sites using Deep Site server.** (DOCX)

**S1 Dataset.** (XLSX)

**S1 File. Processed protein FASTA sequence.** (DOCX)

## Acknowledgments

The authors are grateful to the Department of Biomedical Engineering, MIST for providing support to successfully conduct this research.

## Author Contributions

**Conceptualization:** Tasnia Nabi, Tanver Hasan Riyed, Akid Ornob.

**Data curation:** Tasnia Nabi, Tanver Hasan Riyed.

**Formal analysis:** Tasnia Nabi, Tanver Hasan Riyed, Akid Ornob.

**Investigation:** Tasnia Nabi, Tanver Hasan Riyed, Akid Ornob.

**Methodology:** Tasnia Nabi, Tanver Hasan Riyed.

**Project administration:** Akid Ornob.

**Resources:** Tasnia Nabi, Akid Ornob.

**Software:** Tanver Hasan Riyed.

**Supervision:** Akid Ornob.

**Validation:** Tasnia Nabi, Tanver Hasan Riyed, Akid Ornob.

**Visualization:** Tasnia Nabi, Tanver Hasan Riyed, Akid Ornob.

**Writing – original draft:** Tasnia Nabi, Tanver Hasan Riyed, Akid Ornob.

**Writing – review & editing:** Tasnia Nabi, Tanver Hasan Riyed, Akid Ornob.

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
