## [Decision Letter · Decision Letter 0]

18 Jul 2024

PONE-D-24-18002Deep learning based predictive modeling to screen natural compounds against TNF-alpha for the potential management of Rheumatoid Arthritis: Virtual screening to comprehensive in silico investigationPLOS ONE

Dear Dr. Ornob,

Thank you for submitting your manuscript to PLOS ONE. After careful consideration, we feel that it has merit but does not fully meet PLOS ONE’s publication criteria as it currently stands. Therefore, we invite you to submit a revised version of the manuscript that addresses the points raised during the review process.

We look forward to receiving your revised manuscript.

Kind regards,

Sadiq Umar

Academic Editor

PLOS ONE

Journal Requirements:

Additional Editor Comments (if provided):

Reviewers' comments:

Reviewer's Responses to Questions

**Comments to the Author**

1. Is the manuscript technically sound, and do the data support the conclusions?

Reviewer #1: Yes

2. Has the statistical analysis been performed appropriately and rigorously? 

Reviewer #1: Yes

3. Have the authors made all data underlying the findings in their manuscript fully available?

Reviewer #1: No

4. Is the manuscript presented in an intelligible fashion and written in standard English?

Reviewer #1: Yes

5. Review Comments to the Author

Reviewer #1: The manuscript analyzes a deep learning (DL) approach to finding natural TNF-alpha inhibitors, a potential treatment for rheumatoid arthritis (RA). The authors use a DL model trained on known TNF-alpha inhibitors to predict the bioactivity of natural compounds in the Selleckchem database. The most promising candidates are then analyzed using in silico methods to assess drug-likeness, binding affinity, and stability. This DL method for virtually screening natural compounds against TNF-alpha is a promising strategy for drug discovery. The analysis is strengthened by using a well-established database and incorporating multifaceted in silico analyses. The manuscript is well-organized and provides a detailed explanation of the methodology. However, there are some methodological weaknesses:

Targeting only TNF-alpha may be insufficient for a complex disease like RA. The authors should justify why the model can address broader aspects of the disease.

The manuscript doesn't adequately explain why 2AZ5 (structure of TNF-alpha with a small molecule inhibitor) and CASTp/Deepsite were chosen for site identification in the virtual screening. If the active site features identified by the RCSB PDB structure (2AZ5) were compared to those identified by CASTPA/deepsite, the results should be presented and discussed.

A major concern is the apparent absence of validation for the virtual screening protocol, especially since TNF-alpha is a well-studied target. Without prospective validation, the authors should consider retrospective studies, as suggested in the following references, to gain a better understanding of the protocol's validity: doi/10.1021/jm300687e for dude selection and https://doi.org/10.1039/C8RA09318K for method execution

6. PLOS authors have the option to publish the peer review history of their article (what does this mean?). If published, this will include your full peer review and any attached files.

Reviewer #1: **Yes: **Shafi Ullah Khan

---

## [Author Response · Author response to Decision Letter 0]

20 Sep 2024

Comment 1: 

Targeting only TNF-alpha may be insufficient for a complex disease like RA. The authors should justify why the model can address broader aspects of the disease.

Response 1: 

We thank the reviewer for the thoughtful comment and acknowledge that the pathogenesis of Rheumatoid Arthritis (RA) involves a complex and multifactorial network of various cytokines, cells, and immune pathways (1). We have chosen TNF-alpha (TNFα) as our model protein since it is involved multi-directionally in the pathogenesis of RA and represents the largest cytokine target for RA therapeutics (2,3). Secreted by Th1 cells and macrophages, TNFα was initially thought to play a synergistic role to enhance the destructive behaviour of IL-1 (4). However, follow up studies found that excessive activation of TNF-α signaling led to arthritis even in the absence of functional T and B cells (5). Later studies reported that even the membrane-bound form of TNFα (mTNFα) can lead to the full expression of arthritis (6). Additionally, synovial fibroblasts, activated by TNFα, release other pro-inflammatory cytokines such as IL-6, IL-1β which further accelerates cartilage and bone erosion (7). By targeting TNFα, our model indirectly modulates these interconnected pathways, potentially leading to broader therapeutic effects against RA. 

Comment 2: 

The manuscript doesn't adequately explain why 2AZ5 (structure of TNF-alpha with a small molecule inhibitor) and CASTp/Deepsite were chosen for site identification in the virtual screening. If the active site features identified by the RCSB PDB structure (2AZ5) were compared to those identified by CASTPA/deepsite, the results should be presented and discussed.

Response 2: 

We thank the reviewer for his valuable comments and for making this interesting observation. The 2AZ5 structure of the TNFα protein has been extensively studied in literature for different virtual screening studies (8). The structure is complexed with a small molecule inhibitor and has become a benchmark for drug discovery studies aiming to inhibit the inflammatory activity of the TNFα protein. We report missing residues in the 2AZ5 structure which was filled via homology modeling in Swiss-Model (9). Briefly, the 2AZ5 target protein was uploaded on the Swiss-Model web server and a template search was initiated which identified 2E7A as the highest ranked template based on sequence coverage, similarity, and accuracy of the predicted structure. The resulting model had GMQE score of 0.87, QMEANDisCO (Global) score of 0.82±0.05, and Ramachandran outlier of 0.46%, confirming the validity of its structure. 

The CASTp webserver was used to predict several binding pockets and cavities in our processed protein structure (10). Deepsite extended this capability by using deep convolutional neural networks to forecast which of these pockets are more likely to interact with ligands, therefore offering a functional viewpoint on the geometrically detected sites (11). Interestingly, the most probable binding sites in our model overlapped with those found in the 2E7A structure. These common residues were GLY59, CYS60, PRO91, CYS92, GLN93, ARG94, THR96, ALA100, GLU110, PRO104, and TRP105 in chain A and PRO61, HIS64, SER90, PRO91, CYS92, and GLN93 in chain B. Some of these sites also appear in the 3rd largest binding pocket in the 2AZ5 structure - CYS92 in chain A and CYS60, PRO91, CYS92, GLN93, and ARG94 in chain B. We do not see shared active site features identified in the RCSB PDB structure in our model possibly due to subtle changes in geometric and topological features during homology modeling. Prior work involving 2AZ5 protein structure have identified even more distinct active site features. Saddala and Huang (12) reported Val91, Asn92, Leu93, and Phe124 in chain A and His15, Val17, Ala18, Pro20, Arg32 Ala33, Asn34, Ala35, Phe144, Glu146, Ser147, Gly148, Gln149 and Val150 in chain B as potential binding sites. These sites belong to the 5th largest binding pocket (by volume) of 2AZ5 as identified by the CASTp webserver underscoring the fact that protein preparation steps may lead to differential active site prediction. 

Furthermore, the molecular interaction studies (Fig 7) reveal the multiple and diverse binding between our top-ranked hits and the target protein in the active site regions. Notably, veratramine, which exhibits the highest interaction density, show the lowest binding free energy (ΔGbind = -54.91 kcal/mol) in MM/GBSA analysis. 

Note: When analyzing active sites, it is important to note that there is a shift of 9 residues between our model and the 2AZ5, 2E7A structures as these proteins start from residue 10 (S1 Fig). We re-indexed our protein sequence to start from 1 during the protein preparation step. Therefore, GLY59 reported in our work will correspond to GLY68 in the 2E7A structure. Conversely, PRO100 in the 2E7A structure will correspond to PRO91 in our protein model.

Comment 3: 

A major concern is the apparent absence of validation for the virtual screening protocol, especially since TNF-alpha is a well-studied target. Without prospective validation, the authors should consider retrospective studies, as suggested in the following references, to gain a better understanding of the protocol's validity: doi/10.1021/jm300687e for dude selection and https://doi.org/10.1039/C8RA09318K for method execution

Response 3:

Once again, we thank the reviewer for his insightful feedback and agree that retrospective validation is required to confirm the effectiveness of our virtual screening protocol. The referenced computational tools suggested by the reviewer are widely used in developing virtual screening protocols. However, due to budgetary limitations and time constraints associated with obtaining unpaid licences for these tools, we opted to use open-source alternatives. These open-source solutions are widely accepted, well-validated, and maintain the integrity of our validation process (13). 

Molecular fingerprints have been used in recent literature to validate virtual ligand screening methods and have achieved similar or even superior results compared to 3D shape-based methods for many of the DUD targets (14,15) Given this, we validated our virtual screening using both Morgan and Layered fingerprint descriptors. The Morgan fingerprint (MF) and LayeredFingerprint (16) was calculated using the RDKit Python package. MF had a (radius = 2 and a fingerprint size =2048) setting. The LayeredFingerprint was computed with a minimum path length of 1, a maximum path length of 9, and a fingerprint size of 2048. For benchmarking, we generated and utilized 25 decoy molecules for each of the top 20 active compounds from the DUD-E (17,18) database, and included Schisantherin A (19), a known natural TNF-alpha inhibitor, as a control drug. The anti-inflammatory activity of Schisantherin A against TNF-alpha is well-documented in both computational and in-vitro studies (20). ROC curve analysis showed an AUC of 0.90 for Morgan fingerprints, demonstrating strong predictive power, and 0.80 for Layered fingerprints, indicating reasonable performance as depicted in Fig 1.

Fig 1. Retrospective validation of our virtual screening protocol. a) ROC plot based on Tanimoto scores for MF and Layered fingerprint descriptors b) Heat map showing Tanimoto similarity between control (index 1), actives (index 2-21) and decoys (22-50) and c) Box plots with distribution of Tanimoto scores between active-control and decoy-control for both the fingerprint descriptors

The Enrichment Factor at 1% was 24.81 for Morgan and 12.40 for Layered, highlighting the model’s ability to prioritize active compounds over decoys. The calculated Enrichment Factor (EF) at 1% for binding affinity was 18.18. Considering our limitations in using paid software and that the ROC-AUC metric may not always be optimal for evaluating binding affinity or docking scores in virtual screening, we used EF at top 1% to assess the performance of our model in prioritizating top-ranking compounds (21). A similarity analysis using Tanimoto coefficients compared active compounds and decoys with Schisantherin A. For clarity, we visualized the first 50 compounds, starting with Schisantherin A as the control, followed by 20 active compounds and 29 decoys. The heatmap reveals a clear distinction between the control, active, and decoy compounds. The control compound (index 1) shows notable similarity with many of the active compounds (indices 2-21), while the active compounds themselves form clusters of moderate-to-high Tanimoto similarity, indicating shared structural features. In contrast, the decoys (indices 22-50) generally display lower similarity to both the control and active compounds, with only a few exceptions. Overall, the demarcations in the heat map indicate that active compounds are more similar to each other and the control, while the decoys remain largely dissimilar, validating the effectiveness of the virtual screening process. The boxplots demonstrate statistically significant differences in similarity values between active and decoy compounds for both Morgan (p <10-5) and Layered fingerprint (p <10-6) techniques. These results highlight that active compounds consistently show greater similarity than decoys, validating the effectiveness of the virtual screening process.

References

1. Kondo N, Kuroda T, Kobayashi D. Cytokine Networks in the Pathogenesis of Rheumatoid Arthritis. Int J Mol Sci. 2021 Oct 10;22(20):10922. 

2. Jang D in, Lee AH, Shin HY, Song HR, Park JH, Kang TB, et al. The Role of Tumor Necrosis Factor Alpha (TNF-α) in Autoimmune Disease and Current TNF-α Inhibitors in Therapeutics. Int J Mol Sci. 2021 Mar 8;22(5):2719. 

3. Choy EHS, Panayi GS. Cytokine Pathways and Joint Inflammation in Rheumatoid Arthritis. New England Journal of Medicine. 2001 Mar 22;344(12):907–16. 

4. van de Loo AA, van den Berg WB. Effects of murine recombinant interleukin 1 on synovial joints in mice: measurement of patellar cartilage metabolism and joint inflammation. Ann Rheum Dis. 1990 Apr 1;49(4):238–45. 

5. Keffer J, Probert L, Cazlaris H, Georgopoulos S, Kaslaris E, Kioussis D, et al. Transgenic mice expressing human tumour necrosis factor: a predictive genetic model of arthritis. EMBO J. 1991 Dec;10(13):4025–31. 

6. Georgopoulos S, Plows D, Kollias G. Transmembrane TNF is sufficient to induce localized tissue toxicity and chronic inflammatory arthritis in transgenic mice. J Inflamm. 1996;46(2):86–97. 

7. McInnes IB, Schett G. Pathogenetic insights from the treatment of rheumatoid arthritis. The Lancet. 2017 Jun;389(10086):2328–37. 

8. Parves MdR, Mahmud S, Riza YM, Sujon KM, Uddin MAR, Chowdhury MdIA, et al. Inhibition of TNF-Alpha Using Plant-Derived Small Molecules for Treatment of Inflammation-Mediated Diseases. In: The 1st International Electronic Conference on Biomolecules: Natural and Bio-Inspired Therapeutics for Human Diseases. Basel Switzerland: MDPI; 2020. p. 13. 

9. Waterhouse A, Bertoni M, Bienert S, Studer G, Tauriello G, Gumienny R, et al. SWISS-MODEL: homology modelling of protein structures and complexes. Nucleic Acids Res. 2018 Jul 2;46(W1):W296–303. 

10. Binkowski TA. CASTp: Computed Atlas of Surface Topography of proteins. Nucleic Acids Res. 2003 Jul 1;31(13):3352–5. 

11. Jiménez J, Doerr S, Martínez-Rosell G, Rose AS, De Fabritiis G. DeepSite: protein-binding site predictor using 3D-convolutional neural networks. Bioinformatics. 2017 Oct 1;33(19):3036–42. 

12. Saddala MS, Huang H. Identification of novel inhibitors for TNFα, TNFR1 and TNFα-TNFR1 complex using pharmacophore-based approaches. J Transl Med. 2019 Dec 2;17(1):215. 

13. Bento AP, Hersey A, Félix E, Landrum G, Gaulton A, Atkinson F, et al. An open source chemical structure curation pipeline using RDKit. J Cheminform. 2020 Dec 1;12(1):51. 

14. Cereto-Massagué A, Ojeda MJ, Valls C, Mulero M, Garcia-Vallvé S, Pujadas G. Molecular fingerprint similarity search in virtual screening. Methods. 2015 Jan;71:58–63. 

15. Zhou H, Skolnick J. Utility of the Morgan Fingerprint in Structure-Based Virtual Ligand Screening. J Phys Chem B. 2024 Jun 6;128(22):5363–70. 

16. Pattanaik L, Coley CW. Molecular Representation: Going Long on Fingerprints. Chem. 2020 Jun;6(6):1204–7. 

17. Khan SU, Ahemad N, Chuah LH, Naidu R, Htar TT. Sequential ligand- and structure-based virtual screening approach for the identification of potential G protein-coupled estrogen receptor-1 (GPER-1) modulators. RSC Adv. 2019;9(5):2525–38. 

18. Ni B, Wang H, Khalaf HKS, Blay V, Houston DR. AutoDock-SS: AutoDock for Multiconformational Ligand-Based Virtual Screening. J Chem Inf Model. 2024 May 13;64(9):3779–89. 

19. Boyenle ID, Adelusi TI, Ogunlana AT, Oluwabusola RA, Ibrahim NO, Tolulope A, et al. Consensus scoring-based virtual screening and molecular dynamics simulation of some TNF-alpha inhibitors. Inform Med Unlocked. 2022;28:100833. 

20. Ci X, Ren R, Xu K, Li H, Yu Q, Song Y, et al. Schisantherin A Exhibits Anti-inflammatory Properties by Down-Regulating NF-κB and MAPK Signaling Pathways in Lipopolysaccharide-Treated RAW 264.7 Cells. Inflammation. 2010 Apr 14;33(2):126–36. 

21. Wójcikowski M, Ballester PJ, Siedlecki P. Performance of machine-learning scoring functions in structure-based virtual screening. Sci Rep. 2017 Apr 25;7(1):46710.

---

## [Editor Report · Decision Letter 1]

3 Oct 2024

Deep learning based predictive modeling to screen natural compounds against TNF-alpha for the potential management of Rheumatoid Arthritis: Virtual screening to comprehensive in silico investigation

PONE-D-24-18002R1

Dear Dr. Ornob,

We’re pleased to inform you that your manuscript has been judged scientifically suitable for publication and will be formally accepted for publication once it meets all outstanding technical requirements.

Kind regards,

Sadiq Umar

Academic Editor

PLOS ONE

---

## [Editor Report · Acceptance letter]

14 Oct 2024

PONE-D-24-18002R1 

PLOS ONE

Dear Dr. Ornob, 

I'm pleased to inform you that your manuscript has been deemed suitable for publication in PLOS ONE. Congratulations! Your manuscript is now being handed over to our production team.

Kind regards, 

on behalf of

Dr. Sadiq Umar 

Academic Editor

PLOS ONE